# FGBench: A Dataset and Benchmark for Molecular Property Reasoning at Functional Group-Level in Large Language Models

**Xuan Liu**[1]    **Siru Ouyang**[2]    **Xianrui Zhong**[2]    **Jiawei Han**[2]    **Huimin Zhao**[1]

[1] Department of Chemical and Biomolecular Engineering, University of Illinois Urbana-Champaign
[2] Department of Computer Science, University of Illinois Urbana-Champaign
`{xliu254, siruo2, xzhong23, hanj, zhao5}@illinois.edu`

## Abstract

Large language models (LLMs) have gained significant attention in chemistry. However, most existing datasets center on molecular-level property prediction and overlook the role of fine-grained functional group (FG) information. Incorporating FG-level data can provide valuable prior knowledge that links molecular structures with textual descriptions, which can be used to build more interpretable, structure-aware LLMs for reasoning on molecule-related tasks. Moreover, LLMs can learn from such fine-grained information to uncover hidden relationships between specific functional groups and molecular properties, thereby advancing molecular design and drug discovery. Here, we introduce FGBench, a dataset comprising 625K molecular property reasoning problems with functional group information. Functional groups are precisely annotated and localized within the molecule, which ensures the dataset's interoperability, thereby facilitating further multimodal applications. FGBench includes both regression and classification tasks on 245 different functional groups across three categories for molecular property reasoning: (1) single functional group impacts, (2) multiple functional group interactions, and (3) direct molecular comparisons. In the benchmark of state-of-the-art LLMs on 7K curated data, the results indicate that current LLMs struggle with FG-level property reasoning, highlighting the need to enhance reasoning capabilities in LLMs for chemistry tasks. We anticipate that the methodology employed in FGBench to construct datasets with functional group-level information will serve as a foundational framework for generating new question–answer pairs, enabling LLMs to better understand fine-grained molecular structure–property relationships. The dataset and evaluation code are available at https://github.com/xuanliugit/FGBench.

## 1   Introduction

Large Language Models (LLMs) have become increasingly popular in the chemistry domain, with applications in areas like molecular property prediction [Liu et al., 2024, Jablonka et al., 2024, Qian et al., 2023], molecule captioning [Li et al., 2024, Edwards et al., 2022], and molecule generation [Ye et al., 2025]. However, most existing work concentrates primarily on molecule-level predictions (Figure 1a). This is due to the nature of current databases that typically only provide molecule-level labels, such as MoleculeNet [Wu et al., 2018] and CHEMBL [Gaulton et al., 2012]. Although molecule caption datasets like PubChem [Kim et al., 2023] offer some functional group information associated with molecules, the data available is often ambiguous and incomplete.

Functional groups (FGs) are specific groups of atoms within molecules that impart unique physical and chemical properties [Hanson, 2001]. Scientists summarize these properties by identifying and naming

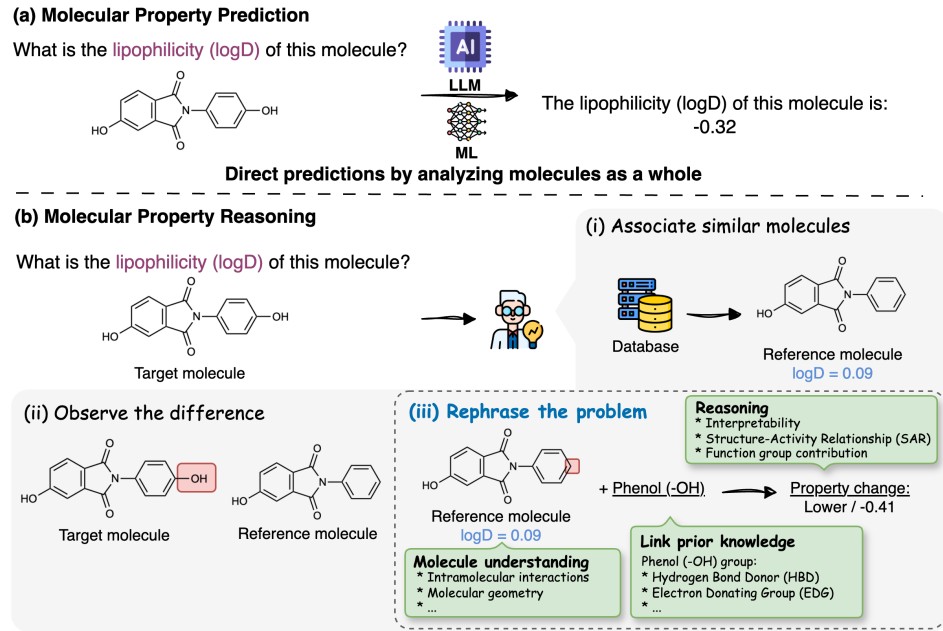

Figure 1: Illustration of LLMs for two molecular property prediction tasks, (a) molecular property prediction and (b) molecular property reasoning.

these atom groups to better understand molecular behavior and interactions. For example, hydroxyl groups (-OH) are known for their polarity and ability to form hydrogen bonds, while carboxylic groups (-COOH) play a critical role in numerous organic reactions, e.g., esterification and thioesterification. Functional groups thus serve as valuable and transferable knowledge, facilitating the reasoning of relationships between molecular structures and their properties. Leveraging this knowledge holds great potential to substantially enhance LLMs by improving their understanding, prediction accuracy, and interpretability of molecular properties, as well as other molecular manipulation tasks, such as synthesis planning [Yu et al., 2023, Liu et al., 2025a].

Recently, there has been growing interest in incorporating functional group-level information into molecular representation pretraining [Li et al., 2023, Nguyen et al., 2024, Sun et al., 2021]. Other studies utilizing molecular fragments in pretraining have also achieved notable results in molecular representation [Luong and Singh, 2023]. Although these studies primarily treat functional groups as tokens for learning molecular representations, their results underscore the importance of detailed, fine-grained molecular information in chemistry-related tasks.

While existing corpora contain extensive descriptions of functional groups and their chemical roles, current molecular property databases lack explicit links between functional groups and molecular properties. This disconnect prevents LLMs from leveraging the rich, fine-grained chemical knowledge embedded in textual data. Bridging this gap by constructing a functional group-centered dataset would enable more interpretable and property-aware LLMs, ultimately advancing their reasoning capabilities in chemistry-related tasks.

Reasoning in chemistry with LLMs is particularly challenging, as it demands a deep understanding of molecular structures and their relationships with specific properties (Figure 1b). When scientists try to give a reasonable prediction on the properties of a target molecule, they often rely on functional groups and similarities to known molecules, following a three-step process: **(i) Associate similar molecules**, where scientists identify similar molecules from databases to retrieve their properties; **(ii) Observe the difference**, where they note functional group differences between the target and reference molecules; and **(iii) Rephrase the problem**, where they infer properties of the target molecule using prior knowledge of functional groups and the reference molecule's overall structure. The third step is crucial for property reasoning, as functional groups provide an important theoretical basis to study the structure-activity relationship (SAR).

However, constructing datasets that support this type of functional group–driven reasoning (step iii) remains a substantial challenge. It requires accurately annotating functional groups in the molecules and identifying the functional group difference between two molecules. Moreover, molecule asymmetry, two-dimensional structural information, and isomerism must be carefully considered. For example, 1-propanol ($CH_3CH_2CH_2OH$) and 2-propanol (($CH_3$)$_2CHOH$) have the same hydroxyl (-OH) functional group, but they have different properties due to different positions of this functional group. Consequently, existing annotation methods used in previous studies [Li et al., 2023, Nguyen et al., 2024, Sun et al., 2021] are insufficient for these tasks, as their inherent annotation schemes prevent them from providing accurate and comprehensive annotations of diverse functional groups [Liu et al., 2025b].

To better explore and enhance the reasoning capabilities of LLMs in chemistry, we introduce FGBench, a novel dataset designed specifically for molecular property reasoning at the functional group level. FGBench contains 625K molecular property reasoning problems across eight different molecular properties, each accompanied by detailed functional group information, including precise positional data. The tasks in FGBench are organized into three dimensions: (1) single functional group impact, (2) multiple functional group interactions, and (3) molecular comparisons. Within each dimension, two categories of question-answer (QA) pairs are provided: Boolean and value-based. Boolean QA pairs assess the model's ability to recognize trends associated with changes in functional groups, while value-based QA pairs evaluate the model's capability to predict exact quantitative changes. Each QA pair includes clear, detailed instructions for editing molecules at the functional group level.

To build FGBench, we developed a new data processing pipeline incorporating a validation-by-reconstruction strategy, which can be generalized to other molecular property datasets to ensure high-quality molecular comparisons. Furthermore, we benchmarked six state-of-the-art open-source and closed-source LLMs using a selected subset of 7K data points from FGBench. Our benchmarking provides valuable insights into the current strengths and limitations of chemical reasoning in LLMs, highlighting critical areas for improvement.

In summary, our contributions are:

- A novel data processing pipeline using a validation-by-reconstruction strategy for reliable functional group-level molecular comparisons, broadly applicable to various molecular datasets.

- FGBench, a dataset of 625K molecular property QA pairs with detailed functional group annotations and precise positions, facilitating advanced LLM fine-tuning, chemical reasoning tasks, and SAR analyses. To our knowledge, FGBench is the first dataset explicitly targeting functional group-level molecular property reasoning.

- A benchmarking study of state-of-the-art LLMs on a 7K subset of FGBench, highlighting current limitations in functional group-level property reasoning and emphasizing the need for improved chemical reasoning in LLMs.

## 2   Related Work

**Molecular Property Dataset.** The development of datasets and benchmarks has significantly advanced molecular property prediction in LLMs, with MoleculeNet [Wu et al., 2018] serving as one of the earliest and most widely used datasets. MoleculeNet provides a comprehensive collection of benchmark datasets covering various molecular tasks, including quantum mechanics, physical chemistry, biophysics, and physiology, facilitating comparative assessment of models across diverse properties. Currently, most question-answer datasets are built on the basis of MoleculeNet. For example, Mol-Instructions [Fang et al., 2024] is an instruction dataset designed for the biomolecular domain, and its molecular property prediction task uses QM9 dataset of MoleculeNet. Similarly, SMolInstruct [Yu et al., 2024] includes 14 selected chemistry tasks for instruction tuning, 6 of which are property prediction tasks. Currently, none of the existing datasets incorporates functional group information to build molecular property reasoning QA.

**Benchmarks for Chemistry.** Chemical datasets are expected to further explore the reasoning capabilities of LLMs by providing either calculation-related questions [Ouyang et al., 2024] or more fine-grained auxiliary information [Guo et al., 2024]. For example, SciBench [Wang et al., 2024] includes college-level chemistry problems with mathematical reasoning. MolPuzzle [Guo et al.,

2024] introduces a molecular structure elucidation problem, which involves deducing a molecule's structure from spectral data with three sub-tasks. For other datasets in chemistry, some of them focus on a general understanding of chemical knowledge, and some of them only focus on molecules. ChemBench [Mirza et al., 2024] comprises over 2,700 question-answer pairs sourced from various chemical disciplines, enabling a rigorous assessment of LLM performance in chemistry. In molecule description tasks, the ChEBI-20 dataset [Edwards et al., 2021] derived from PubChem [Kim et al., 2023] is an important source that provides a wealth of textual descriptions on molecules, including their structures, properties, and biological activities. MoleculeQA [Lu et al., 2024] comprises approximately 62,000 QA pairs covering over 23,000 molecules. Each QA pair consists of a manually crafted question including one correct answer and three distractors, which are all derived from authoritative molecular descriptions.

**Functional Groups.** Functional groups have been widely used in molecular representation learning [Li et al., 2023, Nguyen et al., 2024, Sun et al., 2021]. Most of the existing functional group annotation methods rely on direct pattern matching between molecules and functional groups, like CheckMol [Haider, 2010]. However, these methods fail when there are two functional groups that overlap. In addition, they cannot directly tell the functional group difference between two molecules. AccFG [Liu et al., 2025b] solves these problems, but it still needs a further process to verify the results on the functional group difference between two molecules at the atom level. Other works [Degen et al., 2008, Ghersi and Singh, 2014, Liu et al., 2017, Diao et al., 2023, Yang et al., 2024] decompose molecules into small fragments. Fragmentation provides convenience for tokenizing molecules, but it cannot effectively connect molecular fragments and natural semantics.

# 3 FGBench: Task and Construction

## 3.1 Problem Definition

We formalize the problem of predicting molecular property changes due to the addition or deletion of functional groups as follows: Given a molecule and a functional group modification pair $(M, FG)$, where $M$ is the original molecule and $FG$ is the functional group to be added or deleted. The task aims to reason the change in molecular property $P$ after the modification. The task can be represented as $(M, FG) \rightarrow \Delta P$, where $\Delta P$ can be either a Boolean value (e.g., if change to inactivate from activate) or a numerical value of the property change (e.g., the difference in solubility).

## 3.2 Dataset Construction

The FGBench dataset is constructed from ten existing molecular property datasets from MoleculeNet [Wu et al., 2018], including ESOL, Lipophilicity, FreeSolv, HIV, BACE, BBBP, Tox21, SIDER, ClinTox, and QM9. The complete dataset and tasks used in FGBench are shown in Appendix A.2. The illustration of the dataset construction workflow is shown in Figure 2. For a molecular property dataset $(M, P)$, we first canonicalize the SMILES of the molecules. To lower the calculation cost on the FG difference, we first build a similarity matrix of the molecules based on the Tanimoto similarity of 512-bit Morgan fingerprints [Rogers and Hahn, 2010], and only keep the pairs of molecules $(M_1, M_2)$ with similarity larger than 0.7. Then we process the molecular pairs with AccFG [Liu et al., 2025b] to get the FG difference with accurate positions $(FG_1, FG_2)$. The definition of functional groups is inherently vague and context-dependent, and here we adopt the most informative functional group descriptors as described by Liu et al. [2025b] in this work (see Appendix A.1).

The resulting FG difference is then validated by the strategy of validation-by-reconstruction. This strategy involves removing the FG from the original molecule and replacing it with the FG from the second molecule. The reconstruction process ensures that the modified molecule maintains its structural integrity and is chemically valid. This part also provides the necessary information to build the QA pairs, including (i) molecule SMILES with atom number, (ii) functional group SMILES with atom number, and (iii) connecting positions of the molecule for functional groups. After the validation, the molecular comparison dataset $[(M_1, M_2), (FG_1, FG_2), (P_1, P_2)]$ of 42,967 molecular pairs is generated, which includes the molecular pair, the FG difference, and the properties of each molecule. The molecular comparison dataset covers 245 different functional groups and 27 different alkane lengths. The top 10 most frequent functional groups and alkanes are listed in Table 1.

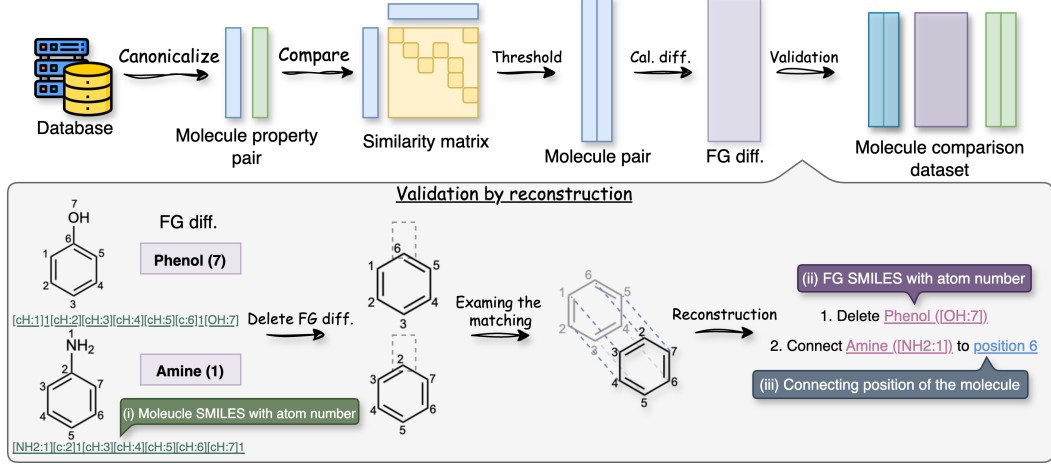

Figure 2: FGBench dataset construction workflow. Molecules from a database are canonicalized and compared to generate a similarity matrix. Molecule pairs exceeding a similarity threshold are selected, and their FG differences are calculated. These differences are validated through a reconstruction process that involves removing and replacing FGs, ensuring structural consistency. The final output is a curated molecule comparison dataset.

Table 1: Functional Group and Alkane Count and Percentage

| Rank | FG name | Count | Percentage | Alkane name | Count | Percentage |
|------|---------|-------|------------|-------------|-------|------------|
| 1 | Hydroxy | 29115 | 19.04 | C1 alkane | 99903 | 80.05 |
| 2 | Alkyne | 13029 | 8.52 | C2 alkane | 13250 | 10.62 |
| 3 | Nitrile | 12716 | 8.32 | C3 alkane | 6014 | 4.82 |
| 4 | Tetrahydrofuran | 6853 | 4.48 | C4 alkane | 2004 | 1.61 |
| 5 | Ether | 6809 | 4.45 | C5 alkane | 930 | 0.75 |
| 6 | Oxetane | 6408 | 4.19 | C6 alkane | 858 | 0.69 |
| 7 | Benzene | 5158 | 3.37 | C7 alkane | 438 | 0.35 |
| 8 | Tetrahydropyran | 4980 | 3.26 | C8 alkane | 421 | 0.34 |
| 9 | Oxirane | 4771 | 3.12 | C10 alkane | 362 | 0.29 |
| 10 | Arylchloride | 4639 | 3.03 | C9 alkane | 199 | 0.16 |

## 3.3 Reasoning Task Categories and QA Pairs

To study various scenarios of functional group-level reasoning, we categorize the tasks into three dimensions: (1) single functional group impact, (2) multiple functional group interactions, and (3) molecular comparisons. Each dimension is further divided into two categories: Boolean and value-based. The Boolean category focuses on whether the property changes after the modification, while the value-based category aims to predict the exact change in property values. The task categories and their illustrations are shown in Table 2.

**Dimension 1: Single functional group impact** filters the dataset to include only one functional group difference. This dimension is designed to assess the model's ability to recognize the impact of a single type of functional group on a base molecule on molecular properties.

**Dimension 2: Multiple functional group interactions** focus on the interaction between multiple functional groups. This dimension is designed to evaluate the model's ability to understand how multiple functional groups interact and influence molecular properties.

**Dimension 3: Molecular comparison** focuses on the overall comparison between two molecules. This dimension is designed to evaluate the model's ability to compare two molecules and understand the differences in their properties without functional group information.

Table 2: FGBench task category and illustration.

| Dimension | Query | Category | Illustration |
|---|---|---|---|
| Single FG impact | Mol + FG | Boolean | $\pm$ FG [loc.] $\xrightarrow{\text{Property?}}$ Increase/decrease (R) Change/Unchange (C) |
| | | Value | $\pm$ FG [loc.] $\xrightarrow{\text{Property?}}$ +0.04/-0.05 (R) |
| Multiple FG interaction | Mol + FGs | Boolean | $\pm$ FG1 [loc.] $\pm$ FG2 [loc.]... $\xrightarrow{\text{Property?}}$ Increase/decrease (R) Change/Unchange (C) |
| | | Value | $\pm$ FG1 [loc.] $\pm$ FG2 [loc.]... $\xrightarrow{\text{Property?}}$ +0.04/-0.05 (R) |
| Molecular comparison | Mol vs Mol | Boolean | vs $\xrightarrow{\text{Property?}}$ Increase/decrease (R) Change/Unchange (C) |
| | | Value | vs $\xrightarrow{\text{Property?}}$ +0.04/-0.05 (R) |

The QA dataset is generated based on these three dimensions through templates as shown in the Appendix A.3. For a molecular comparison pair $[(M_1, M_2), (FG_1, FG_2), (P_1, P_2)]$, the question includes the descriptions of molecule SMILES with atom number $(M_1)$, the property name, and the ground truth value of the property $(P_1)$. Then it describes how to modify the molecule by removing FGs $(FG_1)$ and adding FGs $(FG_2)$ on the original molecule $(M_1)$. This will lead to a new molecule $(M_2)$ with property $(P_2)$, but this information is not included in the question. Then the question asks (i) if the property of the new molecule $(M_2)$ is different from the original molecule $(M_1)$ for classification tasks, (ii) if the property of the new molecule $(M_2)$ is larger than the original molecule $(M_1)$, or what is the value change. An example of the QA pair is shown in Figure 3. The number of molecular pairs and QA pairs in each dimension and category is shown in Table 3. The final FGBench dataset contains a total of 625,936 QA pairs.

To evaluate the performance of LLMs on FGBench, we select a maximum of 25 QA pairs from each task in the Single and the Interaction categories for the benchmark. In order to better compare and analyze within a dataset, we select the same pairs that were used in the Single and the Interaction categories for the Comparison category. This selection ensures a balanced representation across all datasets and task categories, facilitating a comprehensive evaluation of the models' reasoning capabilities. The selected subset for experiment in the next step comprises 7,146 QA pairs, distributed as shown in Table 3.

## 4 Benchmark Large Language Models on FGBench

### 4.1 Experimental Setup

We benchmark the following models on 7,146 test QA pairs: (1) GPT-4o [OpenAI], (2) o3-mini [OpenAI, 2025], (3) Llama-3.1 8B [AI], (4) Llama-3.1 70B [AI], (5) Qwen2.5-7B [Qwen et al., 2025], (6) ChemLLM-7B [Zhang et al., 2024], (7) nach0 [Livne et al., 2024], (8) Llama-3-8B-MolInst[1] [Fang et al., 2024], and (9) LlaSMol-Mistral-7B[2] [Yu et al., 2024]. In the models mentioned, GPT-4o and o3-mini are proprietary (closed-source), while the others are open-source alternatives. Additionally, models (6) to (9) are specifically designed or fine-tuned for chemistry-related tasks. The models are evaluated on the selected subset of FGBench, and the results are reported in terms of accuracy (ACC) for classification tasks (Boolean label) and root mean square error (RMSE) for regression

---

[1]Llama-3-8B-Instruct further tuned with Mol-Instructions
[2]Mistral-7B-v0.1 further tuned with SMolInstruct

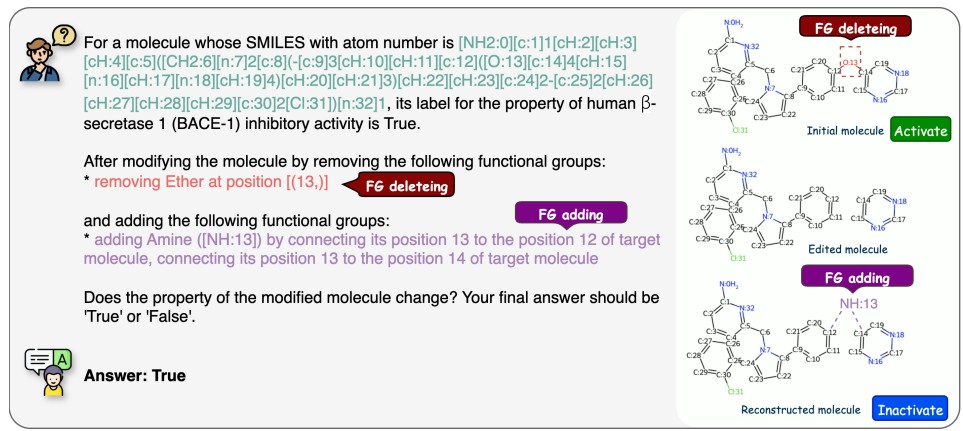

Figure 3: An example of multiple FG interactions from the BACE database. The ether group is deleted from the initial molecule, and an amine group is attached to it. The change related to these two functional groups causes the change of the molecule's BACE-1 inhibitory activity from active to inactive.

Table 3: Summary of FGBench dataset and the selected dataset for benchmark.

| Type | Physical Chemistry | | | Biophysics | | Physiology | | | | Quantum | Total |
|---|---|---|---|---|---|---|---|---|---|---|---|
| Dataset Name | ESOL | Lipo. | FreeSolv | HIV | BACE | BBBP | Tox21 | SIDER | ClinTox | QM9 | |
| # of tasks | 1 | 1 | 1 | 1 | 1 | 1 | 12 | 27 | 1 | 12 | 58 |
| # of molecules | 1,117 | 4,200 | 642 | 41,120 | 1,513 | 1,975 | 7,823 | 1,427 | 1,461 | 131,480 | 192,758 |
| *Dimension*[1] | | | | | | | | | | | |
| Single | 271 | 914 | 132 | 5,051 | 857 | 167 | 1,320 | 48 | 64 | 4,896 | 13,720 |
| Inter. | 42 | 2,792 | 9 | 14,874 | 4,500 | 305 | 1,244 | 121 | 115 | 5,245 | 29,247 |
| Comp. pairs | 313 | 3,706 | 141 | 19,925 | 5,357 | 472 | 2,564 | 169 | 179 | 10,141 | **42,967** |
| *Category*[2] | | | | | | | | | | | |
| Boolean QA | 626 | 7,412 | 282 | 39,850 | 10,714 | 944 | 61,536 | 9,126 | 358 | 243,384 | 373,232 |
| Value QA | 626 | 7,412 | 282 | - | - | - | - | - | - | 243,384 | 251,704 |
| # of QA | 1,252 | 14,824 | 564 | 39,850 | 10,714 | 944 | 61,536 | 9,126 | 358 | 486,768 | **625,936** |
| *QA for benchmark* | | | | | | | | | | | |
| Single - Bool | 25 | 25 | 25 | 25 | 25 | 25 | 300 | 648 | 25 | 300 | 1,423 |
| Single - Value | 25 | 25 | 25 | - | - | - | - | - | - | 300 | 375 |
| Inter. - Bool | 21 | 25 | 4 | 25 | 25 | 25 | 300 | 675 | 25 | 300 | 1,425 |
| Inter. - Value | 21 | 25 | 4 | - | - | - | - | - | - | 300 | 350 |
| Comp. - Bool | 46 | 50 | 29 | 50 | 50 | 50 | 600 | 1,323 | 50 | 600 | 2,848 |
| Comp. - Value | 46 | 50 | 29 | - | - | - | - | - | - | 600 | 725 |
| Total | 184 | 200 | 116 | 100 | 100 | 100 | 1,200 | 2,646 | 100 | 2,400 | **7,146** |

[1] The data in *Dimension* is counted in terms of the number of molecular pairs.
[2] The data in *Category* is counted in terms of the number of QA pairs.
  Single: Single FG impact; Inter.: Multiple FG interaction; Comp.: Molecular comparison.

tasks (value label). Note that separate answer parsers are used for nach0, Llama-3-8B-MolInst, and LlaSMol-Mistral-7B to better accommodate their limited instruction-following capabilities.

## 4.2 Main Result

The performance of various models on the FGBench dataset is summarized in Table 4. The results are divided into three categories: (1) single FG impact, (2) multiple FG interaction, and (3) molecular comparison. For each category, we evaluate the models based on their performance in Boolean and value-based tasks. Our key observations are as follows:

**LLMs exhibit limited understanding of functional group–related tasks and face significant challenges in reasoning about interactions among multiple functional groups.** The best-performing model, o3 mini, achieves an accuracy of 0.687 on single FG impact tasks (Boolean), indicating a certain capability in understanding functional group-level information in molecular property predictions. However, the performance of most models significantly declines in tasks involving multiple FG interactions. For example, GPT-4o gets 0.667 accuracy on the Single category, but gets 0.488 on the Interaction category. Similarly, Llama-3.1 70B decreases to 0.53 from 0.683. This highlights a clear

Table 4: Performance of open-source LLMs and closed-source LLMs on FGBench dataset.

| | Single | | | | Interaction | | | | Comparison | | | |
|---|---|---|---|---|---|---|---|---|---|---|---|---|
| | Boolean 1423 | | Value 375 | | Boolean 1425 | | Value 350 | | Boolean 2848 | | Value 725 | |
| # of sample | ACC(↑) | Valid(↑) | RMSE(↓) | Valid(↑) | ACC(↑) | Valid(↑) | RMSE(↓) | Valid(↑) | ACC(↑) | Valid(↑) | RMSE(↓) | Valid(↑) |
| *Closed-source* | | | | | | | | | | | | |
| GPT-4o | 0.667 | 0.999 | 77.990 | 0.813 | 0.488 | 0.998 | 43.577 | 0.891 | 0.614 | 0.992 | 68.857 | 0.708 |
| o3 mini | **0.687** | 0.999 | 101.886 | 0.960 | **0.693** | **1.000** | 39.943 | 0.977 | **0.703** | **1.000** | **64.579** | **0.975** |
| *Open-source* | | | | | | | | | | | | |
| Llama-3.1 8B | 0.548 | 0.993 | 162.351 | 0.840 | 0.547 | 0.982 | 421.325 | 0.780 | 0.474 | 0.991 | 80.566 | 0.761 |
| Llama-3.1 70B | 0.683 | **1.000** | 84.119 | **0.973** | 0.530 | **1.000** | 38.646 | 0.977 | 0.456 | **1.000** | 64.887 | 0.943 |
| Qwen2.5-7B | 0.590 | 0.999 | **63.511** | 0.576 | 0.396 | 0.999 | **36.307** | 0.683 | 0.664 | **1.000** | 65.471 | 0.223 |
| ChemLLM-7B | 0.233 | 0.997 | 209.584 | 0.629 | 0.235 | 0.997 | 162.742 | 0.566 | 0.250 | **1.000** | 65.428 | 0.514 |
| nach0-base[1] | 0.606 | 0.798 | 104.534 | 0.539 | 0.543 | 0.756 | 172.929 | 0.683 | 0.041 | 0.149 | 12221.946 | 0.879 |
| Llama-3-8B-MolInst[1,2] | 0.107 | 0.203 | 328.935 | 0.496 | 0.059 | 0.149 | 188.376 | 0.486 | 0.469 | 0.873 | 138.654 | 0.837 |
| LlaSMol-Mistral-7B[1] | 0.387 | 0.922 | 266.720 | 0.923 | 0.298 | 0.968 | 262.550 | **0.983** | 0.239 | 1.000 | 245.298 | 0.924 |

[1] These results are calculated by separate answer parsers.
[2] The model is trained on SELFIES (SELF-referencing embedded string) as the molecular descriptor.
The overall optimal results are in bold.

difficulty among LLMs in accurately reasoning about interactions between multiple functional groups and their combined effects on molecular properties. It is worth noting that some LLMs, like o3 mini and Qwen2.5-7B, get a better performance in the Comparison category compared with the other two categories. This may contribute to these models having already seen the MoleculeNet dataset that matches the question in the Comparison category, which directly provides the whole molecule to LLMs. However, the fact that Qwen2.5-7B achieves 0.664 accuracy in the Comparison category and 0.396 accuracy in the Interaction category suggests that molecule-level information alone may not be sufficient for the model to learn FG-level knowledge.

**Reasoning models and larger models tend to achieve better performance, but the improvements are still limited.** The reasoning-based models, o3-mini and Qwen2.5-7B, demonstrate improved performance on value-based tasks compared to Boolean tasks. This may be attributed to the reasoning-focused training these models received, providing them with enhanced intuition for quantitative molecular property predictions within FGBench. Note that value-based datasets exhibit different orders of magnitude in their numerical scales. Therefore, it is important to consider both accuracy and validity when making comparisons. For a more equitable comparison across different datasets, please refer to Appendix B.1. Additionally, o3-mini achieves the best performance on 4 out of 6 tasks, which further indicates that reasoning ability is important for property-related tasks. On the other hand, larger models show improved yet still limited performance on these tasks, as evidenced by the comparison between Llama-3.1 8B and Llama-3.1 70B.

**Current LLMs fine-tuned for chemistry tasks lack generalizability to unseen but closely related chemistry tasks.** ChemLLM-7B, despite having a similar size to Llama-3.1 8B, underperforms on FGBench significantly. Although ChemLLM-7B is trained extensively on molecule-level data, it still exhibits poor performance on FGBench tasks. This indicates that model architecture and training data also critically influence reasoning performance. In addition, other chemistry-specific models, like nach0, Llama-3-8B-MolInst, and LlaSmol-Mistral-7B, also show results that are inferior to other general models, even if they have been trained on the original MoleculeNet dataset. These models present a limited instruction following ability, so separate parsers are used to extract their answers. We observe that nach0-base achieves 0.606 accuracy on the Single-Boolean task, which is significantly higher than other chemistry-specific models. However, a further examination of the results reveals that this outcome is largely driven by the bias of nach0-base, which predicted "False" for 97.7% (1073 out of 1098) of the samples in the Single Boolean task. These underscore that relying solely on molecule-level datasets and previous post-training strategies is inadequate for enhancing functional group-level understanding and reasoning in LLMs.

In addition, we observe that some models' performance on molecular comparison is better than the other two reasoning tasks. This might be attributed to these models already being trained on the molecular dataset. However, this performance is suspected to be widely generalized to more out-of-distribution molecules like FGs.

## 4.3 Failure Analysis

Figure 4 illustrates an example of a reasoning error made by the o3-mini model when answering a single functional group (FG) impact question from the Lipophilicity dataset. The task asks whether

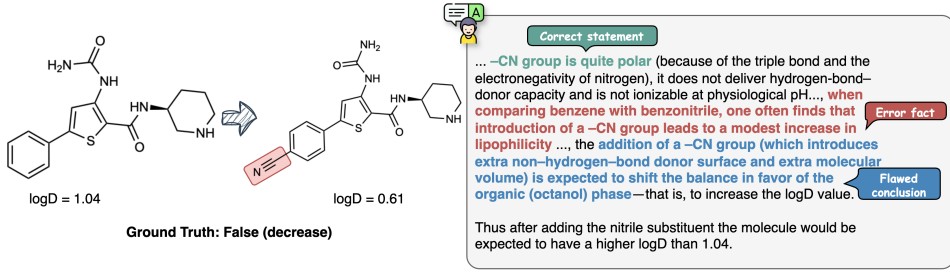

Figure 4: An example output from o3-mini on a single FG impact QA (Boolean) from Lipophilicity dataset. Left is the original molecule. Right is the molecule after adding a nitrile (-CN) group. The o3-mini response is in the gray box.

the addition of a nitrile (–CN) group to a given molecule (left) increases its logD value. While the ground truth indicates a decrease in logD (from 1.04 to 0.61), the model incorrectly predicts an increase. Notably, the model demonstrates partially correct reasoning by identifying that the nitrile group is polar and non-ionizable at physiological pH, correctly recognizing that it does not contribute hydrogen-bond donor character. However, the model fails in factual interpretation and reasoning transfer. The specific case (benzonitrile vs. benzene) provided by it has the same FG difference (a nitrile group). The model says benzonitrile has a higher logD than benzene, but the fact is that benzene has a higher logD. This flawed example highlights the model's limitations in contextual reasoning and transferability across molecular scaffolds. This example suggests that integrating retrieval mechanisms could enhance the model's performance. Although the model recalled a relevant case (benzonitrile), it failed to correctly contextualize its applicability. This case suggests that a retrieval-augmented approach, providing similar examples with structural and physicochemical context, could help ground the model's reasoning in more appropriate analogies and improve prediction accuracy for property shifts caused by functional group modifications. The full output in Figure 4 and more examples can be found in Appendix B.2.

## 5   Conclusion and Outlook

In this work, we introduced FGBench, a novel dataset designed to evaluate and enhance the reasoning capabilities of LLMs in chemistry, specifically, molecular property prediction at the functional group level. FGBench provides a comprehensive framework for molecular property reasoning, encompassing single functional group impacts, multiple functional group interactions, and molecular comparisons. Our benchmarking results highlight the challenges faced by current LLMs in understanding and reasoning about functional group-level information, emphasizing the need for further advancements in this area.

**Development of structure-aware reasoning LLMs for chemistry.** Molecular property reasoning relies on the principle "structure determines properties". Hence, the development of structure-aware reasoning LLMs is quite important for understanding the semantic descriptions of molecule structures (e.g., functional groups and their multiplication on molecules) and understanding molecule language itself (e.g., SMILES, SELFIES, or a new modality for molecules). Additionally, explicit reasoning and latent reasoning for studying structure–property relationships are two open directions. FGBench provides a perspective that mimicking human heuristics, such as reasoning via functional groups, offers critical interpretability and verifiability advantages. The latent reasoning in chemistry is also promising, but it has yet to be explored.

**Development of new training strategies for chemistry-specific LLMs.** While existing chemistry-focused LLMs perform well on familiar tasks, their poor results on FGBench reveal the limited generalizability of chemical knowledge. In addition, directly fine-tuning LLMs on large chemistry-related corpora may distort the base model's original general capabilities and chemical understanding capabilities. Thus, new training strategies are needed to empower base models with transferable chemical knowledge while mitigating catastrophic forgetting.

Looking forward, FGBench opens up several promising research directions for advancing LLMs in chemistry. First, the dataset's fine-grained annotations, including explicit functional group identities and positions, provide a valuable foundation for developing LLMs with a deeper and more detailed understanding of molecular structures and their properties. Second, the structure of FGBench, centered around functional group-level comparisons and reasoning, can drive the development of more capable and interpretable reasoning models to make property predictions based on structural differences. Third, the dataset's alignment with molecular graphs and structural information makes it well-suited for multi-modal learning, encouraging the development of multimodal LLMs that integrate textual, graphical, and 3D molecular representations. Together, these directions can lead to more intelligent and chemistry-aware language models for applications in molecular design, drug discovery, and beyond. We anticipate that FGBench will not only advance the development of LLMs in chemistry but also inspire new methodologies for multi-modal reasoning and editing in scientific domains.

# 6 Limitation

While FGBench represents a significant step toward enabling functional group-level molecular property reasoning in large language models (LLMs), several limitations remain that point to promising directions for future research. Although our dataset includes detailed annotations of functional groups (FGs) and their precise positions within molecules, the current framework primarily focuses on FG identity and FG-level isomerism (e.g., distinguishing between isopropyl and propyl groups). However, more fine-grained structural distinctions, such as position isomerism (e.g., ortho-, meta-, para- substitution in aromatics), carbon chain isomerism (e.g., linear vs. branched alkanes), and stereoisomerism (e.g., enantiomers and diastereomers), are not explicitly considered in this version of the dataset. These structural variations may lead to significantly different physicochemical or pharmacological properties and are critical for more advanced structure–activity relationship (SAR) reasoning. Incorporating these forms of isomerism would require additional stereochemical annotations and more sophisticated graph-based or 3D-aware functional group extraction algorithms, which we leave for future development.

## Acknowledgments and Disclosure of Funding

This work was supported by the IBM-Illinois Discovery Accelerator Institute, the Molecule Maker Lab Institute: An AI Research Institutes program supported by the US National Science Foundation (NSF) under grant no. 2019897, and the National Science Foundation IIS-19-56151.

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

# A  FGBench construction details

## A.1  The definition of functional groups

This work uses the same definition and the same annotation principle for functional groups as described in AccFG [Liu et al., 2025b], which uses the most informative functional group descriptors in the vocabulary. For example, describing a -COOH group as "carboxylate" conveys more precise information than decomposing it into "hydroxy" and "ketone," which would dilute the semantic specificity. AccFG prioritizes functional groups with higher informativeness, such as "Triazene", over less informative alternatives like "Amine" or "Azo," thereby enabling more compact and meaningful molecular descriptions.

## A.2  Complete dataset and tasks used in FGBench

The complete dataset and tasks used in FGBench are shown in Table 5

## A.3  Question templates used to build QA in FGBench

**System Instruction:**

> You are an expert in chemistry and molecular science.

**Overall Instruction:**

> You are given a problem related to molecular property. Conclude the answer by stating "The answer is therefore \boxed{[ANSWER]}".

### A.3.1  Single Molecule Questions

**Boolean Classification:**

> For a molecule whose SMILES with atom number is {target_mapped_smiles}, its label for the property of {property_name} is {target_label}.
> After modifying the molecule {edit_text}
> Does the property of the modified molecule change? Your final answer should be 'True' or 'False'.

**Boolean Regression:**

> For a molecule whose SMILES with atom number is {target_mapped_smiles}, its label for the property of {property_name} is {target_label}.
> After modifying the molecule {edit_text}
> Does the property of the modified molecule increase? Your final answer should be 'True' or 'False'.

**Value Regression:**

> For a molecule whose SMILES with atom number is {target_mapped_smiles}, its label for the property of {property_name} is {target_label}.
> After modifying the molecule {edit_text}
> What is the value change of the property for the modified molecule? Your final answer should be "[value]" for increase or "-[value]" for decrease.

### A.3.2  Interaction Questions

**Boolean Classification (Interaction):**

> For a molecule whose SMILES with atom number is {target_mapped_smiles}, its label for the property of {property_name} is {target_label}.

Table 5: Complete dataset and task descriptions used in FGBench

| Dataset Type | Dataset | Task |
|---|---|---|
| Regression | ESOL | Log-scale water solubility (mol/L) |
| Regression | Lipophilicity | Octanol/water distribution coefficient (logD at pH 7.4) |
| Regression | FreeSolv | Hydration free energy in water |
| Classification | HIV | HIV inhibitory activity |
| Classification | BACE | Human $\beta$-secretase 1 (BACE-1) inhibitory activity |
| Classification | BBBP | Blood-brain barrier penetration |
| Classification | ClinTox | Drugs approved by the FDA and passed clinical trials |
| Classification | Tox21 | Androgen receptor pathway activation |
| Classification | Tox21 | Androgen receptor ligand-binding domain activation |
| Classification | Tox21 | Aryl hydrocarbon receptor activation |
| Classification | Tox21 | Inhibition of aromatase enzyme |
| Classification | Tox21 | Estrogen receptor pathway activation |
| Classification | Tox21 | Estrogen receptor ligand-binding domain activation |
| Classification | Tox21 | Activation of PPAR$\gamma$ |
| Classification | Tox21 | Activation of antioxidant response element signaling |
| Classification | Tox21 | Activation of ATAD5-mediated DNA damage response |
| Classification | Tox21 | Activation of heat shock factor response element signaling |
| Classification | Tox21 | Disruption of mitochondrial membrane potential |
| Classification | Tox21 | Activation of p53 tumor suppressor pathway |
| Classification | SIDER | Cause liver and bile system disorders |
| Classification | SIDER | Cause metabolic and nutritional disorders |
| Classification | SIDER | Cause product-related issues |
| Classification | SIDER | Cause eye disorders |
| Classification | SIDER | Cause abnormal medical test results |
| Classification | SIDER | Cause muscle, bone, and connective tissue disorders |
| Classification | SIDER | Cause gastrointestinal disorders |
| Classification | SIDER | Cause adverse social circumstances |
| Classification | SIDER | Cause immune system disorders |
| Classification | SIDER | Cause reproductive system and breast disorders |
| Classification | SIDER | Cause tumors and abnormal growths (benign, malignant, or unspecified) |
| Classification | SIDER | Cause general disorders and administration site conditions |
| Classification | SIDER | Cause endocrine (hormonal) disorders |
| Classification | SIDER | Cause complications from surgical and medical procedures |
| Classification | SIDER | Cause vascular (blood vessel) disorders |
| Classification | SIDER | Cause blood and lymphatic system disorders |
| Classification | SIDER | Cause skin and subcutaneous tissue disorders |
| Classification | SIDER | Cause congenital, familial, and genetic disorders |
| Classification | SIDER | Cause infections and infestations |
| Classification | SIDER | Cause respiratory and chest disorders |
| Classification | SIDER | Cause psychiatric disorders |
| Classification | SIDER | Cause renal and urinary system disorders |
| Classification | SIDER | Cause complications during pregnancy, childbirth, or perinatal period |
| Classification | SIDER | Cause ear and balance disorders |
| Classification | SIDER | Cause cardiac disorders |
| Classification | SIDER | Cause nervous system disorders |
| Classification | SIDER | Cause injury, poisoning, and procedural complications |
| Regression | QM9 | Dipole moment (unit: D) |
| Regression | QM9 | Isotropic polarizability (unit: Bohr$\hat{3}$) |
| Regression | QM9 | Highest occupied molecular orbital energy (unit: Hartree) |
| Regression | QM9 | Lowest unoccupied molecular orbital energy (unit: Hartree) |
| Regression | QM9 | Gap between HOMO and LUMO (unit: Hartree) |
| Regression | QM9 | Electronic spatial extent (unit: Bohr$\hat{2}$) |
| Regression | QM9 | Zero point vibrational energy (unit: Hartree) |
| Regression | QM9 | Heat capavity at 298.15K (unit: cal/(mol*K)) |
| Regression | QM9 | Internal energy at 0K (unit: Hartree) |
| Regression | QM9 | Internal energy at 298.15K (unit: Hartree) |
| Regression | QM9 | Enthalpy at 298.15K (unit: Hartree) |
| Regression | QM9 | Free energy at 298.15K (unit: Hartree) |

After modifying the molecule {edit_text}

Does the property of the modified molecule change? Your final answer should be 'True' or 'False'.

**Boolean Regression (Interaction):**

For a molecule whose SMILES with atom number is {target_mapped_smiles}, its label for the property of {property_name} is {target_label}.

After modifying the molecule {edit_text}

Does the property of the modified molecule increase? Your final answer should be 'True' or 'False'.

**Value Regression (Interaction):**

For a molecule whose SMILES with atom number is {target_mapped_smiles}, its label for the property of {property_name} is {target_label}.

After modifying the molecule {edit_text}

What is the value change of {property_name} for the modified molecule? Your final answer should be "[value]" for increase or "-[value]" for decrease.

### A.3.3 Comparison Questions

**Boolean Classification (Comparison):**

For a target molecule whose SMILES is {target_smiles} and a reference molecule whose SMILES is {ref_smiles}, the reference molecule has the label for the property of {property_name} as {ref_label}. Does the target molecule have a different property label compared to the reference molecule? Your final answer should be 'True' or 'False'.

**Boolean Regression (Comparison):**

For a target molecule whose SMILES is {target_smiles} and a reference molecule whose SMILES is {ref_smiles}, the reference molecule has the label for the property of {property_name} as {ref_label}. Does the target molecule have a higher value of the property compared to the reference molecule? Your final answer should be 'True' or 'False'.

**Value Regression (Comparison):**

For a target molecule whose SMILES is {target_smiles} and a reference molecule whose SMILES is {ref_smiles}, the reference molecule has the value for the property of {property_name} as {ref_label}. What is the value change of {property_name} for the target molecule compared to the reference molecule? Your final answer should be "[value]" for increase or "-[value]" for decrease.

# B  Evaluation Experiments

## B.1  Result in each dataset

Table 6: GPT-4o result on different dataset

| | Single | | | | Interaction | | | | Comparison | | | |
|---|---|---|---|---|---|---|---|---|---|---|---|---|
| | Boolean | | Value | | Boolean | | Value | | Boolean | | Value | |
| | ACC(↑) | Valid(↑) | RMSE(↓) | Valid(↑) | ACC(↑) | Valid(↑) | RMSE(↓) | Valid(↑) | ACC(↑) | Valid(↑) | RMSE(↓) | Valid(↑) |
| esol | 1.000 | 1.000 | 0.725 | 0.960 | 0.667 | 1.000 | 1.046 | 0.857 | 0.870 | 1.000 | 1.050 | 0.978 |
| lipo | 0.840 | 1.000 | 0.484 | 1.000 | 0.840 | 1.000 | 1.303 | 1.000 | 0.700 | 0.960 | 0.826 | 0.980 |
| freesolv | 0.440 | 1.000 | 0.937 | 0.920 | 0.250 | 1.000 | 1.549 | 0.750 | 0.517 | 1.000 | 0.905 | 0.897 |
| qm9 | 0.463 | 0.993 | 89.229 | 0.777 | 0.537 | 0.990 | 47.192 | 0.887 | 0.510 | 0.993 | 78.669 | 0.655 |
| hiv | 0.840 | 1.000 | - | - | 0.520 | 1.000 | - | - | 0.780 | 1.000 | - | - |
| bace | 0.800 | 1.000 | - | - | 0.440 | 1.000 | - | - | 0.780 | 1.000 | - | - |
| bbbp | 0.560 | 1.000 | - | - | 0.600 | 1.000 | - | - | 0.740 | 1.000 | - | - |
| tox21 | 0.920 | 1.000 | - | - | 0.510 | 1.000 | - | - | 0.703 | 0.990 | - | - |
| sider | 0.628 | 1.000 | - | - | 0.434 | 1.000 | - | - | 0.597 | 0.992 | - | - |
| clintox | 0.600 | 1.000 | - | - | 0.560 | 1.000 | - | - | 0.500 | 1.000 | - | - |

Table 7: o3 mini result on different dataset

| | Single | | | | Interaction | | | | Comparison | | | |
|---|---|---|---|---|---|---|---|---|---|---|---|---|
| | Boolean | | Value | | Boolean | | Value | | Boolean | | Value | |
| | ACC(↑) | Valid(↑) | RMSE(↓) | Valid(↑) | ACC(↑) | Valid(↑) | RMSE(↓) | Valid(↑) | ACC(↑) | Valid(↑) | RMSE(↓) | Valid(↑) |
| esol | 1.000 | 1.000 | 1.433 | 0.960 | 0.857 | 1.000 | 0.829 | 1.000 | 0.891 | 1.000 | 0.969 | 0.978 |
| lipo | 0.840 | 1.000 | 0.436 | 1.000 | 0.760 | 1.000 | 0.868 | 1.000 | 0.760 | 1.000 | 0.713 | 1.000 |
| freesolv | 1.000 | 1.000 | 0.400 | 1.000 | 0.750 | 1.000 | 1.727 | 0.750 | 0.862 | 1.000 | 0.837 | 0.966 |
| qm9 | 0.493 | 1.000 | 114.309 | 0.953 | 0.473 | 1.000 | 43.152 | 0.977 | 0.528 | 1.000 | 71.054 | 0.973 |
| hiv | 0.800 | 1.000 | - | - | 0.640 | 1.000 | - | - | 0.360 | 1.000 | - | - |
| bace | 0.840 | 1.000 | - | - | 0.520 | 1.000 | - | - | 0.620 | 1.000 | - | - |
| bbbp | 0.640 | 1.000 | - | - | 0.680 | 1.000 | - | - | 0.760 | 1.000 | - | - |
| tox21 | 0.950 | 1.000 | - | - | 0.787 | 1.000 | - | - | 0.823 | 1.000 | - | - |
| sider | 0.617 | 0.998 | - | - | 0.744 | 1.000 | - | - | 0.723 | 1.000 | - | - |
| clintox | 0.680 | 1.000 | - | - | 0.880 | 1.000 | - | - | 0.880 | 1.000 | - | - |

Table 8: Llama-3.1 8B result on different dataset

| | Single | | | | Interaction | | | | Comparison | | | |
|---|---|---|---|---|---|---|---|---|---|---|---|---|
| | Boolean | | Value | | Boolean | | Value | | Boolean | | Value | |
| | ACC(↑) | Valid(↑) | RMSE(↓) | Valid(↑) | ACC(↑) | Valid(↑) | RMSE(↓) | Valid(↑) | ACC(↑) | Valid(↑) | RMSE(↓) | Valid(↑) |
| esol | 0.960 | 1.000 | 2.784 | 0.920 | 0.571 | 1.000 | 1.791 | 0.619 | 0.696 | 1.000 | 2.044 | 0.870 |
| lipo | 0.800 | 1.000 | 0.902 | 0.720 | 0.560 | 1.000 | 1.620 | 0.840 | 0.600 | 1.000 | 1.177 | 0.880 |
| freesolv | 0.600 | 0.960 | 1.473 | 0.920 | 0.750 | 1.000 | 0.880 | 0.500 | 0.517 | 0.966 | 0.540 | 0.690 |
| qm9 | 0.480 | 0.993 | 181.872 | 0.837 | 0.517 | 1.000 | 452.193 | 0.790 | 0.500 | 0.983 | 89.427 | 0.747 |
| hiv | 0.800 | 1.000 | - | - | 0.680 | 1.000 | - | - | 0.380 | 1.000 | - | - |
| bace | 0.520 | 1.000 | - | - | 0.440 | 0.920 | - | - | 0.420 | 1.000 | - | - |
| bbbp | 0.360 | 1.000 | - | - | 0.440 | 1.000 | - | - | 0.400 | 1.000 | - | - |
| tox21 | 0.900 | 0.990 | - | - | 0.733 | 0.990 | - | - | 0.542 | 0.992 | - | - |
| sider | 0.403 | 0.994 | - | - | 0.487 | 0.969 | - | - | 0.429 | 0.993 | - | - |
| clintox | 0.160 | 1.000 | - | - | 0.280 | 1.000 | - | - | 0.420 | 0.960 | - | - |

Table 9: Llama-3.1 70B result on different dataset

| | Single | | | | Interaction | | | | Comparison | | | |
|---|---|---|---|---|---|---|---|---|---|---|---|---|
| | Boolean | | Value | | Boolean | | Value | | Boolean | | Value | |
| | ACC(↑) | Valid(↑) | RMSE(↓) | Valid(↑) | ACC(↑) | Valid(↑) | RMSE(↓) | Valid(↑) | ACC(↑) | Valid(↑) | RMSE(↓) | Valid(↑) |
| esol | 1.000 | 1.000 | 1.181 | 0.960 | 0.571 | 1.000 | 1.210 | 1.000 | 0.848 | 1.000 | 1.056 | 0.957 |
| lipo | 0.800 | 1.000 | 0.552 | 0.920 | 0.840 | 1.000 | 1.517 | 0.960 | 0.540 | 1.000 | 0.893 | 0.920 |
| freesolv | 0.920 | 1.000 | 0.641 | 0.960 | 0.250 | 1.000 | 0.632 | 1.000 | 0.724 | 1.000 | 0.489 | 0.966 |
| qm9 | 0.503 | 1.000 | 93.726 | 0.980 | 0.517 | 1.000 | 41.749 | 0.977 | 0.533 | 1.000 | 71.330 | 0.943 |
| hiv | 0.720 | 1.000 | - | - | 0.240 | 1.000 | - | - | 0.240 | 1.000 | - | - |
| bace | 0.640 | 1.000 | - | - | 0.280 | 1.000 | - | - | 0.440 | 1.000 | - | - |
| bbbp | 0.600 | 1.000 | - | - | 0.600 | 1.000 | - | - | 0.580 | 1.000 | - | - |
| tox21 | 0.900 | 1.000 | - | - | 0.480 | 1.000 | - | - | 0.402 | 1.000 | - | - |
| sider | 0.648 | 1.000 | - | - | 0.569 | 1.000 | - | - | 0.435 | 1.000 | - | - |
| clintox | 0.560 | 1.000 | - | - | 0.400 | 1.000 | - | - | 0.260 | 1.000 | - | - |

Table 10: ChemLLM-7B result on different dataset

| | Single | | | | Interaction | | | | Comparison | | | |
|---|---|---|---|---|---|---|---|---|---|---|---|---|
| | Boolean | | Value | | Boolean | | Value | | Boolean | | Value | |
| | ACC(↑) | Valid(↑) | RMSE(↓) | Valid(↑) | ACC(↑) | Valid(↑) | RMSE(↓) | Valid(↑) | ACC(↑) | Valid(↑) | RMSE(↓) | Valid(↑) |
| esol | 0.400 | 1.000 | 5.319 | 0.800 | 0.619 | 0.905 | 3.932 | 0.905 | 0.565 | 1.000 | 2.602 | 0.717 |
| lipo | 0.480 | 0.840 | 1.129 | 0.480 | 0.520 | 1.000 | 0.968 | 0.600 | 0.460 | 1.000 | 1.459 | 0.440 |
| freesolv | 0.600 | 1.000 | 0.639 | 0.520 | 0.500 | 1.000 | 1.130 | 0.750 | 0.414 | 1.000 | 0.317 | 0.345 |
| qm9 | 0.450 | 1.000 | 232.962 | 0.637 | 0.510 | 1.000 | 180.471 | 0.537 | 0.515 | 1.000 | 71.995 | 0.513 |
| hiv | 0.160 | 1.000 | - | - | 0.040 | 1.000 | - | - | 0.100 | 1.000 | - | - |
| bace | 0.120 | 1.000 | - | - | 0.200 | 1.000 | - | - | 0.200 | 1.000 | - | - |
| bbbp | 0.040 | 1.000 | - | - | 0.160 | 1.000 | - | - | 0.100 | 1.000 | - | - |
| tox21 | 0.050 | 1.000 | - | - | 0.083 | 1.000 | - | - | 0.067 | 1.000 | - | - |
| sider | 0.208 | 1.000 | - | - | 0.175 | 0.997 | - | - | 0.208 | 1.000 | - | - |
| clintox | 0.040 | 1.000 | - | - | 0.040 | 1.000 | - | - | 0.140 | 1.000 | - | - |

Table 11: Qwen2.5-7B result on different dataset

| | Single | | | | Interaction | | | | Comparison | | | |
|---|---|---|---|---|---|---|---|---|---|---|---|---|
| | Boolean | | Value | | Boolean | | Value | | Boolean | | Value | |
| | ACC(↑) | Valid(↑) | RMSE(↓) | Valid(↑) | ACC(↑) | Valid(↑) | RMSE(↓) | Valid(↑) | ACC(↑) | Valid(↑) | RMSE(↓) | Valid(↑) |
| esol | 1.000 | 1.000 | 2.460 | 0.920 | 0.714 | 1.000 | 1.162 | 0.905 | 0.674 | 1.000 | 3.693 | 0.478 |
| lipo | 0.800 | 1.000 | 0.701 | 0.640 | 0.640 | 1.000 | 1.417 | 0.720 | 0.540 | 1.000 | 2.773 | 0.200 |
| freesolv | 0.320 | 1.000 | 0.419 | 0.720 | 0.500 | 1.000 | 1.628 | 0.500 | 0.690 | 1.000 | 0.416 | 0.414 |
| qm9 | 0.487 | 0.993 | 74.018 | 0.530 | 0.453 | 0.993 | 39.685 | 0.667 | 0.473 | 1.000 | 76.691 | 0.197 |
| hiv | 0.600 | 1.000 | - | - | 0.040 | 1.000 | - | - | 0.620 | 1.000 | - | - |
| bace | 0.280 | 1.000 | - | - | 0.240 | 1.000 | - | - | 0.600 | 1.000 | - | - |
| bbbp | 0.320 | 1.000 | - | - | 0.280 | 1.000 | - | - | 0.540 | 0.980 | - | - |
| tox21 | 0.753 | 1.000 | - | - | 0.393 | 1.000 | - | - | 0.702 | 1.000 | - | - |
| sider | 0.573 | 1.000 | - | - | 0.385 | 1.000 | - | - | 0.738 | 1.000 | - | - |
| clintox | 0.560 | 1.000 | - | - | 0.160 | 1.000 | - | - | 0.840 | 1.000 | - | - |

Table 12: nach0-base result on different dataset

| | Single | | | | Interaction | | | | Comparison | | | |
|---|---|---|---|---|---|---|---|---|---|---|---|---|
| | Boolean | | Value | | Boolean | | Value | | Boolean | | Value | |
| | ACC(↑) | Valid(↑) | RMSE(↓) | Valid(↑) | ACC(↑) | Valid(↑) | RMSE(↓) | Valid(↑) | ACC(↑) | Valid(↑) | RMSE(↓) | Valid(↑) |
| esol | 0.560 | 0.960 | 1.647 | 0.280 | 0.381 | 0.857 | 2.345 | 0.571 | 0.087 | 0.587 | 2.823 | 0.870 |
| lipo | 0.280 | 0.600 | 1.215 | 0.240 | 0.240 | 0.400 | 2.039 | 0.120 | 0.060 | 0.460 | 2.313 | 1.000 |
| freesolv | 0.440 | 0.960 | 0.476 | 0.560 | 0.500 | 1.000 | 0.784 | 0.500 | 0.103 | 0.517 | 0.914 | 0.828 |
| qm9 | 0.483 | 0.940 | 112.308 | 0.583 | 0.470 | 0.990 | 179.427 | 0.740 | 0.060 | 0.442 | 13488.363 | 0.872 |
| hiv | 0.400 | 0.520 | - | - | 0.600 | 0.600 | - | - | 0.620 | 0.740 | - | - |
| bace | 0.600 | 0.640 | - | - | 0.400 | 0.520 | - | - | 0.020 | 0.020 | - | - |
| bbbp | 0.720 | 0.720 | - | - | 0.640 | 0.760 | - | - | 0.000 | 0.020 | - | - |
| tox21 | 0.767 | 0.813 | - | - | 0.693 | 0.800 | - | - | 0.018 | 0.025 | - | - |
| sider | 0.610 | 0.742 | - | - | 0.523 | 0.659 | - | - | 0.020 | 0.028 | - | - |
| clintox | 0.720 | 0.760 | - | - | 0.600 | 0.640 | - | - | 0.020 | 0.060 | - | - |

Table 13: Llama-3-8B-MolInst result on different dataset

| | Single | | | | Interaction | | | | Comparison | | | |
|---|---|---|---|---|---|---|---|---|---|---|---|---|
| | Boolean | | Value | | Boolean | | Value | | Boolean | | Value | |
| | ACC(↑) | Valid(↑) | RMSE(↓) | Valid(↑) | ACC(↑) | Valid(↑) | RMSE(↓) | Valid(↑) | ACC(↑) | Valid(↑) | RMSE(↓) | Valid(↑) |
| esol | 0.000 | 0.000 | 1.251 | 0.080 | 0.000 | 0.000 | 2.371 | 0.381 | 0.478 | 1.000 | 2.160 | 0.848 |
| lipo | 0.080 | 0.120 | 2.024 | 0.480 | 0.120 | 0.280 | 1.999 | 0.720 | 0.420 | 0.980 | 1.597 | 0.960 |
| freesolv | 0.000 | 0.040 | 0.183 | 0.640 | 0.000 | 0.000 | 0.000 | 0.000 | 0.345 | 0.724 | 0.141 | 0.655 |
| qm9 | 0.020 | 0.030 | 359.173 | 0.520 | 0.073 | 0.157 | 204.675 | 0.480 | 0.455 | 0.848 | 152.617 | 0.835 |
| hiv | 0.040 | 0.240 | - | - | 0.000 | 0.040 | - | - | 0.340 | 0.920 | - | - |
| bace | 0.040 | 0.320 | - | - | 0.000 | 0.120 | - | - | 0.480 | 0.840 | - | - |
| bbbp | 0.080 | 0.320 | - | - | 0.080 | 0.080 | - | - | 0.660 | 0.980 | - | - |
| tox21 | 0.063 | 0.193 | - | - | 0.067 | 0.197 | - | - | 0.273 | 0.808 | - | - |
| sider | 0.181 | 0.292 | - | - | 0.055 | 0.138 | - | - | 0.570 | 0.906 | - | - |
| clintox | 0.160 | 0.280 | - | - | 0.000 | 0.040 | - | - | 0.380 | 0.820 | - | - |

Table 14: LlaSMol-Mistral-7B result on different dataset

| | Single | | | | Interaction | | | | Comparison | | | |
|---|---|---|---|---|---|---|---|---|---|---|---|---|
| | Boolean | | Value | | Boolean | | Value | | Boolean | | Value | |
| | ACC(↑) | Valid(↑) | RMSE(↓) | Valid(↑) | ACC(↑) | Valid(↑) | RMSE(↓) | Valid(↑) | ACC(↑) | Valid(↑) | RMSE(↓) | Valid(↑) |
| esol | 0.400 | 1.000 | 4.007 | 0.680 | 0.286 | 0.952 | 5.191 | 1.000 | 0.478 | 1.000 | 2.650 | 0.978 |
| lipo | 0.520 | 0.960 | 0.647 | 0.400 | 0.520 | 0.960 | 1.402 | 0.800 | 0.460 | 1.000 | 1.095 | 0.980 |
| freesolv | 0.560 | 1.000 | 0.899 | 0.880 | 0.250 | 1.000 | 1.122 | 1.000 | 0.517 | 1.000 | 0.663 | 1.000 |
| qm9 | 0.447 | 0.993 | 287.881 | 0.990 | 0.513 | 1.000 | 281.612 | 0.997 | 0.500 | 1.000 | 271.479 | 0.912 |
| hiv | 0.320 | 1.000 | - | - | 0.080 | 1.000 | - | - | 0.100 | 1.000 | - | - |
| bace | 0.360 | 0.960 | - | - | 0.200 | 1.000 | - | - | 0.160 | 1.000 | - | - |
| bbbp | 0.200 | 0.880 | - | - | 0.160 | 0.840 | - | - | 0.100 | 1.000 | - | - |
| tox21 | 0.590 | 0.933 | - | - | 0.380 | 0.890 | - | - | 0.070 | 1.000 | - | - |
| sider | 0.278 | 0.870 | - | - | 0.184 | 0.991 | - | - | 0.195 | 1.000 | - | - |
| clintox | 0.040 | 1.000 | - | - | 0.040 | 1.000 | - | - | 0.060 | 1.000 | - | - |

## B.2  Failure Examples

### B.2.1  Full result in Figure 4

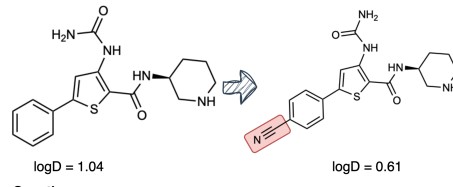

logD = 1.04    logD = 0.61

**Question:**

For a molecule whose SMILES with atom number is [NH2:0][C:1](=[O:2])
[NH:3][c:4]1[cH:5][c:6](-[c:7]2[cH:8][cH:9][cH:10][cH:11][cH:12]2)[s:13]
[c:14]1[C:15](=[O:16])[NH:17][C@H:18]1[CH2:19][CH2:20][CH2:21]
[NH:22][CH2:23]1, its label for the property of octanol/water distribution
coefficient (logD at pH 7.4) is 1.04.

After modifying the molecule by adding the following functional groups:
* adding Nitrile ([N:0]#[C:1]) by connecting its position 1 to the position 10
of target molecule

Does the property of the modified molecule increase? Your final answer
should be 'True' or 'False'.

**o3-mini response:**

We start by noting that the property under consideration is the octanol/water
distribution coefficient (logD at pH 7.4) and that the original molecule has logD =
1.04. In the given modification a nitrile group (–C≈N) is added by connecting it
to an aromatic carbon (position 10 of the target molecule). Although the –CN
group is quite polar (because of the triple bond and the electronegativity of
nitrogen), it does not deliver hydrogen-bond–donor capacity and is not ionizable
at physiological pH. In fragment–based QSAR estimates the nitrile moiety is
typically found to add some hydrophobic character (for example, when
comparing benzene with benzonitrile, one often finds that introduction of a –CN
group leads to a modest increase in lipophilicity or a relatively small change
overall). In the context of a larger druglike molecule that already contains polar
(e.g. amide and amino) groups, the addition of a –CN group (which introduces
extra non–hydrogen–bond donor surface and extra molecular volume) is
expected to shift the balance in favor of the organic (octanol) phase—that is, to
increase the logD value.

Thus after adding the nitrile substituent the molecule would be expected to have
a higher logD than 1.04.

The answer is therefore \boxed{True}.

## B.2.2  Single example

A failure example of o3 mini response on Single category of Lipophilicity:

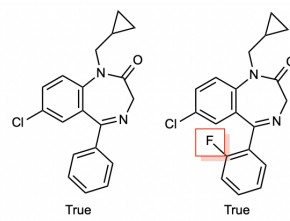

logD = 2.01

logD = 1.97

**Question:**

For a molecule whose SMILES with atom number is [NH$_2$:0][C:1](=[O:2])[c:3]$_1$[cH:4][n:5][c:6]([NH:7][C@H:8]$_2$[CH$_2$:9][CH$_2$:10][CH$_2$:11][NH:12][CH$_2$:13]$_2$)[c:14]$_2$[cH:15][c:16](-[c:17]$_3$[cH:18][cH:19][c:20]([F:21])[cH:22][cH:23]$_3$)[s:24][c:25]$_{12}$, its label for the property of octanol/water distribution coefficient (logD at pH 7.4) is 2.01.

After modifying the molecule by adding the following functional groups:
* adding Arylfluoride ([F:24]) by connecting its position 24 to the position 23 of target molecule

Does the property of the modified molecule increase? Your final answer should be 'True' or 'False'.

**o3 mini response:**

Fluorine atoms on aryl rings tend to increase the hydrophobicity and lipophilicity of the molecule. When an additional Arylfluoride group is introduced, the overall molecule is expected to become more hydrophobic, thus increasing its octanol/water distribution coefficient (logD).

The answer is therefore \boxed{True}.

A failure example of o3 mini response on Single category of BACE:

True

False

**Question:**

For a molecule whose SMILES with atom number is [N:0]#[C:1][c:2]1[cH:3][cH:4][c:5]([C:6](=[O:7])[NH:8][c:9]2[cH:10][cH:11][c:12]([F:13])[c:14]([C@@:15]3([CH2:16][F:17])[N:18]=[C:19]([NH2:20])[O:21][CH2:22][C@@H:23]3[F:24])[cH:25]2)[n:26][cH:27]1, its label for the property of human β-secretase 1 (BACE-1) inhibitory activity is True.

After modifying the molecule by removing the following functional groups:
* removing Fluoride at postion [(17,)]

Does the property of the modified molecule change? Your final answer should be 'True' or 'False'.

**o3 mini response:**

Removing a single fluoride group (at atom position 17) in our molecular framework generally is not expected to alter the overall binding or inhibitory functionality that results in BACE-1 inhibitory activity, which remains True. Therefore, the removal of the fluoride from that peripheral CH2 group does not change the property label.

The answer is therefore \boxed{False}.

A failure example of o3 mini response on Single category of BBBP:

True

True

**Question:**

For a molecule whose SMILES with atom number is [O:0]=[C:1]1[CH2:2][N:3]=[C:4]([c:5]2[cH:6][cH:7][cH:8][cH:9][cH:10]2)[c:11]2[cH:12][c:13]([Cl:14])[cH:15][cH:16][c:17]2[N:18]1[CH2:19][CH:20]1[CH2:21][CH2:22]1, its label for the property of blood-brain barrier penetration is True.

After modifying the molecule by adding the following functional groups:
* adding Arylfluoride ([F:11]) by connecting its position 11 to the position 10 of target molecule

Does the property of the modified molecule change? Your final answer should be 'True' or 'False'.

**o3 mini response:**

The original molecule has been given a property label of True for blood–brain barrier penetration. The attempted modification is the addition of an arylfluoride (–F substituent) that is attached from an aryl position (shifting from atom 11 on the added group to the aryl position 10 on the target molecule). Such a substitution is a relatively minor change; fluorine, being small and only modestly polar (and in an aromatic setting, even less so), typically does not significantly influence the overall properties that determine permeability, such as molecular weight, polar surface area, or lipophilicity in a way that would block BBB penetration. Instead, fluorine substitutions are often used in medicinal chemistry to improve metabolic stability and sometimes even enhance permeability.

Given these factors, the modification is not expected to drastically change the characteristics responsible for blood–brain barrier penetration.

The answer is therefore \boxed{True}.

### B.2.3 Interaction example

A failure example of o3 mini response on Interaction category of Lipophilicity:

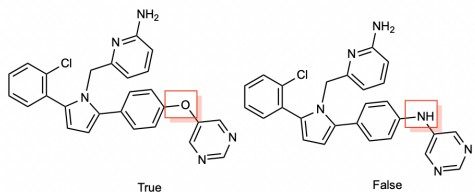

logD = 3.6

logD = 3.2

**o3 mini response:**

Let's analyze the modifications step-by-step. The original molecule has an experimentally determined logD (at pH 7.4) of 3.6, meaning it has a fairly lipophilic character. Two parts are removed and one new group is added:

1. Removing the ether fragment (the –O– at atom 2) eliminates a polar oxygen that engages in hydrogen-bonding and helps solubilize the molecule in water. Therefore, its removal eliminates a somewhat water-affine (and thus "logD-lowering") component. That change tends to increase the overall lipophilicity.

2. Removing the C2 alkyl group (CH3–CH2– corresponding to atoms 0 and 1) takes away a hydrophobic (lipophilic) fragment. On its own, this loss would be expected to decrease lipophilicity (and thus lower logD).

3. Finally, we add an "arylfluoride" substituent by connecting a fluorinated aryl component (here represented by [F:25]) to the aromatic core at position 3. Halogen substitution on aromatic rings, particularly fluorine, usually increases lipophilicity. In addition, attachment of an aryl (rather than a simple alkyl) substituent tends to enhance hydrophobic surface area.

When we balance these effects the removal of the ether (a polar group) tends to push the logD upward, while the loss of a short C2 alkyl chain tends to reduce it. The addition of an arylfluoride, however, not only substitutes one hydrophobic group for the alkyl chain but also introduces an aromatic moiety with a halogen atom, which generally has a stronger logD-enhancing effect than that provided by a simple C2 alkyl group.

Thus, the overall net effect is an increase in the molecule's lipophilicity compared to the original (logD = 3.6).

The answer is therefore \boxed{True}.

**Question:**

For a molecule whose SMILES with atom number is [CH3:0][CH2:1][O:2] [c:3]1[cH:4][cH:5][cH:6][c:7]([NH:8][c:9]2[n:10][n:11][c:12]([C:13](=[O:14])[NH:15] [c:16]3[cH:17][cH:18][c:19]([N:20]4[CH2:21][CH2:22][O:23][CH2:24][CH2:25]4) [n:26]6[cH:27]3[o:28]2)[cH:29]1, its label for the property of octanol/water distribution coefficient (logD at pH 7.4) is 3.6.

After modifying the molecule by removing the following functional groups:
* removing Ether at position [(2,)]
* removing C2 alkane at position [(0, 1)]
and adding the following functional groups:
* adding Arylfluoride ([F:25]) by connecting its position 25 to the position 3 of target molecule

Does the property of the modified molecule increase? Your final answer should be 'True' or 'False'.

A failure example of o3 mini response on Interaction category of BACE:

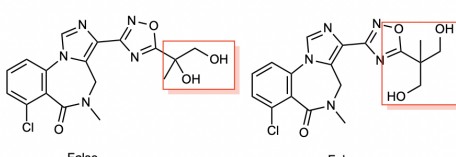

True

False

**Question:**

For a molecule whose SMILES with atom number is [NH2:0][c:1]1[cH:2][cH:3] [cH:4][c:5]([CH2:6][n:7]2[c:8](-[c:9]3[cH:10][cH:11][c:12]([O:13][c:14]4[cH:15][n:16] [cH:17][n:18][cH:19]4)[cH:20][cH:21]3)[cH:22][cH:23][c:24]2-[c:25]2[cH:26][cH:27] [cH:28][cH:29][c:30]2[Cl:31])[n:32]1, its label for the property of human β-secretase 1 (BACE-1) inhibitory activity is True.

After modifying the molecule by removing the following functional groups:
* removing Ether at position [(13,)]
and adding the following functional groups:
* adding Amine ([NH:13]) by connecting its position 13 to the position 12 of target molecule, connecting its position 13 to the position 14 of target molecule

Does the property of the modified molecule change? Your final answer should be 'True' or 'False'.

**o3 mini response:**

We start with a molecule whose BACE-1 inhibitory activity label is True. In the given modification, an ether group (the O-atom at position 13) is removed and an amine group ([NH:13]) is installed connecting position 13 to both positions 12 and 14. Such a replacement—removing an oxygen and "replacing" it with an –NH group that is attached in similar positions—is essentially an isosteric replacement. In medicinal chemistry, it is common to "swap" an ether for an amino substituent in a way that, while it changes details of electronic distribution (for example switching a hydrogen-bond acceptor to one that is also a potential donor), often the overall binding profile remains preserved if the geometry and sterics do not change drastically.

Given that the modification is made in a way that retains the overall molecular framework while "correcting" one of the key functionalities through an isosteric change, the hypothesis is that the BACE-1 inhibitory activity (labeled True for the original molecule) is maintained after the modification.

Thus, the property of the molecule does not change.

The answer is therefore \boxed{False}.

A failure example of o3 mini response on Interaction category of BBBP:

False

False

**Question:**

For a molecule whose SMILES with atom number is [CH3:0][N:1]1[CH2:2] [c:3]2[c:4](-[c:5]3[n:6][o:7][c:8]([C:9]([CH3:10])([OH:11])[CH2:12][OH:13])[n:14]3) [n:15][cH:16][n:17]2-[c:18]2[cH:19][cH:20][cH:21][c:22]([Cl:23]) [c:24]2[C:25]1=[O:26], its label for the property of blood-brain barrier penetration is False.

After modifying the molecule by removing the following functional groups:
* removing 1,2-Diol at position [(11, 9, 12, 13)]
* removing C1 alkane at position [(10,)]
and adding the following functional groups:
* adding C4 alkane ([C:9]([CH3:10])([CH2:11])[CH2:13]) by connecting its position 9 to the position 8 of target molecule, connecting its position 11 to the position 12 of Hydroxy ([OH:12]), connecting its position 13 to the position 14 of Hydroxy ([OH:14])

Does the property of the modified molecule change? Your final answer should be 'True' or 'False'.

**o3 mini response:**

The primary factors that affect blood–brain barrier (BBB) penetration include lipophilicity and the presence of polar groups. In the original molecule, the 1,2-diol is a polar functional group that likely reduces BBB penetration, contributing to the negative (False) label. Removing the 1,2-diol and a small C1 alkane combined with the addition of a more hydrophobic C4 alkane chain increases the overall lipophilicity of the molecule. These modifications are expected to facilitate better passage through the BBB. Thus, the modification alters the original property.

The answer is therefore \boxed{True}.

### B.2.4 Comparison example

A failure example of o3 mini response on Comparison category of Lipophilicity:

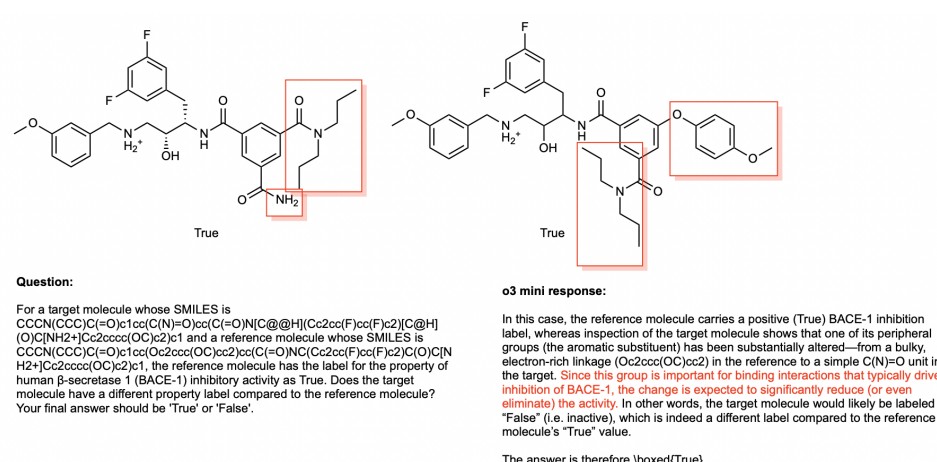

**Question:**

For a target molecule whose SMILES is
NC1(C(=O)NC(CCCN2CCCC2)c2ccc(Cl)cc2)CCN(c2ncnc3[nH]ccc23)CC1 and a
reference molecule whose SMILES is
NC1(C(=O)NC(CO)c2ccc(Cl)cc2)CCN(c2ncnc3[nH]ccc23)CC1, the reference
molecule has the label for the property of octanol/water distribution coefficient
(logD at pH 7.4) as 2.4. Does the target molecule have a higher value of the
property compared to the reference molecule? Your final answer should be 'True'
or 'False'.

**o3 mini response:**

The target molecule has replaced a hydroxymethyl group (–CH2OH) with a
longer, more lipophilic substituent (–CH2CH2CH2–N(cyclic)) that lacks a
hydrogen-bond donating hydroxyl group and introduces an additional nonpolar
cyclic amine. This modification generally increases hydrophobicity and,
consequently, the octanol/water distribution coefficient (logD). Since the
reference molecule has a logD of 2.4, it is reasonable to conclude that the target
molecule will have a higher logD value.

The answer is therefore \boxed{True}.

A failure example of o3 mini response on Comparison category of BACE:

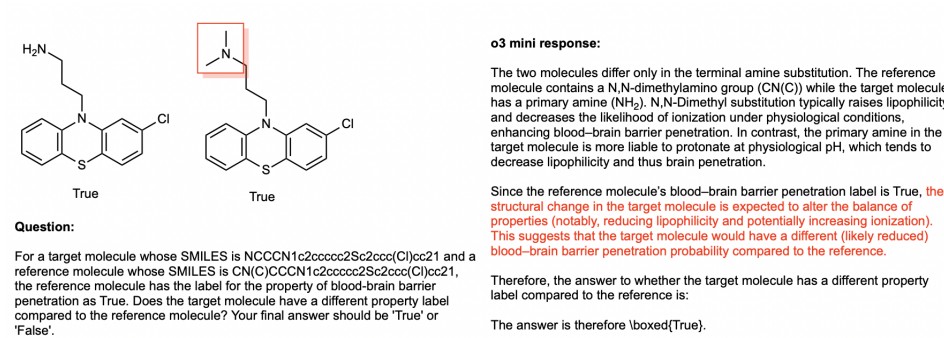

**Question:**

For a target molecule whose SMILES is
CCCN(CCC)C(=O)c1cc(C(N)=O)cc(C(=O)N[C@@H](Cc2cc(F)cc(F)c2)[C@H]
(O)C[NH2+]Cc2cccc(OC)c2)c1 and a reference molecule whose SMILES is
CCCN(CCC)C(=O)c1cc(Oc2ccc(OC)cc2)cc(C(=O)NC(Cc2cc(F)cc(F)c2)C(O)C[N
H2+]Cc2cccc(OC)c2)c1, the reference molecule has the label for the property of
human β-secretase 1 (BACE-1) inhibitory activity as True. Does the target
molecule have a different property label compared to the reference molecule?
Your final answer should be 'True' or 'False'.

**o3 mini response:**

In this case, the reference molecule carries a positive (True) BACE-1 inhibition
label, whereas inspection of the target molecule shows that one of its peripheral
groups (the aromatic substituent) has been substantially altered—from a bulky,
electron-rich linkage (Oc2ccc(OC)cc2) in the reference to a simple C(N)=O unit in
the target. Since this group is important for binding interactions that typically drive
inhibition of BACE-1, the change is expected to significantly reduce (or even
eliminate) the activity. In other words, the target molecule would likely be labeled
"False" (i.e. inactive), which is indeed a different label compared to the reference
molecule's "True" value.

The answer is therefore \boxed{True}.

A failure example of o3 mini response on Comparison category of BBBP:

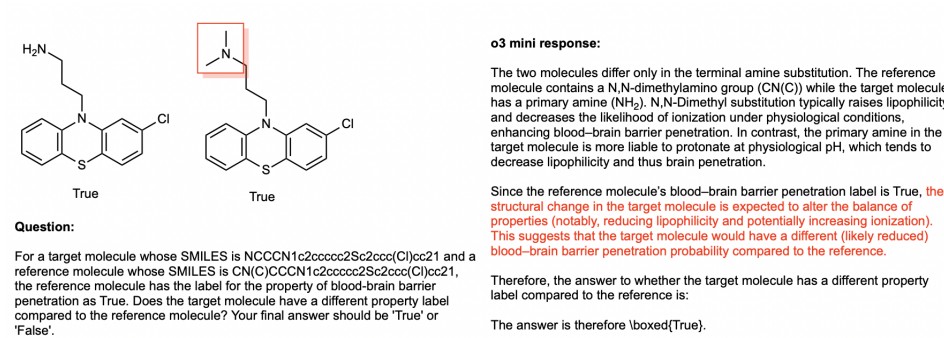

**Question:**

For a target molecule whose SMILES is NCCCN1c2ccccc2Sc2ccc(Cl)cc21 and a
reference molecule whose SMILES is CN(C)CCCN1c2ccccc2Sc2ccc(Cl)cc21,
the reference molecule has the label for the property of blood-brain barrier
penetration as True. Does the target molecule have a different property label
compared to the reference molecule? Your final answer should be 'True' or
'False'.

**o3 mini response:**

The two molecules differ only in the terminal amine substitution. The reference
molecule contains a N,N-dimethylamino group (CN(C)) while the target molecule
has a primary amine (NH$_2$). N,N-Dimethyl substitution typically raises lipophilicity
and decreases the likelihood of ionization under physiological conditions,
enhancing blood–brain barrier penetration. In contrast, the primary amine in the
target molecule is more liable to protonate at physiological pH, which tends to
decrease lipophilicity and thus brain penetration.

Since the reference molecule's blood–brain barrier penetration label is True, the
structural change in the target molecule is expected to alter the balance of
properties (notably, reducing lipophilicity and potentially increasing ionization).
This suggests that the target molecule would have a different (likely reduced)
blood–brain barrier penetration probability compared to the reference.

Therefore, the answer to whether the target molecule has a different property
label compared to the reference is:

The answer is therefore \boxed{True}.

# C Compute Resources

For the FGBench data processing, we employed E5-2683V4 2.1GHz 16-Core Processor with 128G
memory. For the evaluation with LLMs, we employed API calls for GPT-4o and o3-mini. For all
other open-sourced models used in this study, we did the experiments using an NVIDIA A100 GPU
in a high-performance computing unit.

