# OpenReview forum: "FGBench: A Dataset and Benchmark for Molecular Property Reasoning at Functional Group-Level in Large Language Models"
_NeurIPS.cc/2025/Datasets_and_Benchmarks_Track — NeurIPS 2025 Datasets and Benchmarks Track poster_

### Official Review · Reviewer_AxbV · 2025-07-02

**Rating:** 5
**Confidence:** 4

**Summary:**

The paper introduces FGBench, a set of benchmark tasks for molecular property prediction using functional-group-level reasoning. The goal is to evaluate the ability of LLMs to predict how specific changes in the molecular structure (changes in so-called functional groups) impact a given molecular property. Therefore, FGBench contains pairs of structurally similar molecules from different property prediction tasks. Each pair is annotated with the differences in functional groups and the corresponding property differences. Using these pairs, the authors define two major types of tasks: 1) Given molecule A, how does a specified functional group change affect property X? 2) Given molecules A and B, how does property X differ between them? The authors evaluate several open- and closed-source LLMs on these tasks, finding that all models show moderate performance.

**Additional Feedback:**

* How do the authors handle changes in parts of the molecule that are not labeled as “functional groups”?
* “However, more fine-grained structural distinctions, such as position isomerism (e.g., ortho-, meta-, para- substitution in aromatics), carbon chain isomerism (e.g., linear vs. branched alkanes), and stereoisomerism (e.g., enantiomers and diastereomers), are not explicitly considered in this version of the dataset.” This statement is unclear. What exactly do the authors refer to? The current framework is able to represent all kinds of constitutional isomerism, and can e.g. distinguish whether a functional group is attached in the ortho, meta or para position of an arene, or whether a functional group is attached at a linear or a branched position of an alkane backbone. This statement should be revised for clarity.

**Dataset Code Accessibility:**

Yes

**Dataset Code Comments:**

The data could be readily accessed through the `datasets` library.

**Ethical Considerations:**

No, there are no or only very minor ethics concerns

**Final Justification:**

As stated in my original review, I think this paper presents an interesting benchmark on introducing human heuristics (functional groups) into LLM-based reasoning for molecular property prediction. My main concern with this paper was the over-reliance on a heuristic which in itself is vague and context-dependent. If the authors consider this limitation in their manuscript (both in the introduction and in the discussion of the results), I think this is a valid and interesting study. Therefore, I recommend acceptance of the paper (score: 5).

**Limitations Weaknesses:**

As it stands, the paper makes several simplified and potentially misleading statements about functional groups. It is important to clarify that any molecular property always arises from the structure of the entire molecule. Functional groups (FGs), originally introduced for describing reactivity, are a heuristic tool to simplify the complexity of these effects. That said, the concept of what constitutes a FG is itself ambiguous. Any part of a molecule influences the overall properties, and the distinction between a FG and the molecular “backbone” usually depends on the context. In addition, the applicability of FG heuristics for property reasoning is limited. For instance, the effects of FGs are typically context-dependent and non-additive. Factors such as their spatial orientation, interactions with other parts of the molecule, or environmental influences can significantly affect the contribution of a specific FG to the molecular property.

This leads to a broader, perhaps philosophical question: Should LLMs be encouraged to mimic human heuristics, such as the use of functional groups? Or should we allow AI models to identify alternative, possibly more effective patterns in structure–property relationships?

This discussion should not diminish the value of FGBench, which I think is a useful benchmark. However, I believe that the paper needs a more nuanced discussion of the definition and limitations of FGs for molecular property prediction.

**Strengths Contributions:**

**Relevance**: Molecular property/activity prediction is a central challenge in multiple industries. Introducing reasoning abilities this process could improve robustness, interpretability, and trustworthiness of predictive models.

**Clarity**: The paper is well organized, clearly written, and easy to follow.

**Technical Implementation**: The data curation pipeline is carefully designed and well described, and provides high-quality data. Importantly, it does not only contain information on functional group identity, but also on their relative position in the molecule, which is crucial for the prediction of many physical properties. The dataset is a challenging benchmark to evaluate human-level reasoning in molecular property prediction.

---

> ### Author Rebuttal · Authors · 2025-07-31
>
> We thank you for your encouraging feedback and thoughtful review, which prompted us to reflect more deeply on the broader implications of our work. Please find our responses to your questions and concerns below, which we will also implement together within the updated version of the paper.
>
> > Comment #1:
> As it stands, the paper makes several simplified and potentially misleading statements about functional groups. It is important to clarify that any molecular property always arises from the structure of the entire molecule. Functional groups (FGs), originally introduced for describing reactivity, are a heuristic tool to simplify the complexity of these effects. That said, the concept of what constitutes a FG is itself ambiguous. Any part of a molecule influences the overall properties, and the distinction between a FG and the molecular “backbone” usually depends on the context. In addition, the applicability of FG heuristics for property reasoning is limited. For instance, the effects of FGs are typically context-dependent and non-additive. Factors such as their spatial orientation, interactions with other parts of the molecule, or environmental influences can significantly affect the contribution of a specific FG to the molecular property.
>
> Response:
> We thank you for pointing out this. We fully agree that any part of a molecule influences the overall properties. In fact, we view this complexity not as a limitation, but as a central motivation for FGBench. The core task we introduce, predicting property changes based on FG modifications within a molecular context, is explicitly not a simple rule-based process. As the reviewer notes, adding a given FG to different scaffolds can lead to very different outcomes. This underscores the need for models to go beyond memorization and instead perform context-aware reasoning over both the FG and the molecular backbone.
> In our dataset and task design, we do not treat FGs in isolation. Each question provides full structural information, including both the molecular scaffold and the FG modification, to ensure the model is equipped with the necessary context to make nuanced predictions. The goal is to evaluate whether LLMs can use chemically relevant substructures like FGs as reasoning anchors, as human chemists often do, rather than assume they are fixed, universal property determinants.
> Regarding the point that the definition of a functional group can be ambiguous, we agree and believe this makes FGs especially interesting objects of study. FGs are ultimately human-defined constructs representing chemically meaningful atom arrangements. Their semantic flexibility offers two complementary avenues:
>
> 1. Reasoning: leveraging established FG knowledge to make informed deductions about property changes in new molecules.
>
> 2. Discovery: using structure–property patterns in data to uncover new interpretations, contextual effects, or even novel substructures that may warrant recognition as new FGs.
> FGBench is primarily designed to support the first direction of evaluating a model’s ability to reason with existing FG knowledge. However, we anticipate it may also serve as a foundation for the second, by enabling models to learn and generalize new structure–function relationships beyond traditional FG definitions.
>
> > Comment #2
> This leads to a broader, perhaps philosophical question: Should LLMs be encouraged to mimic human heuristics, such as the use of functional groups? Or should we allow AI models to identify alternative, possibly more effective patterns in structure–property relationships?
>
> Response:
> We thank you for this thoughtful comment.  We view this as an open and important question, analogous to the broader debate in the LLM community between explicit (interpretable) reasoning and latent (emergent) reasoning. Both directions are valid and likely complementary.
>
> In the current stage of chemistry-focused LLM research, we believe that mimicking human heuristics, such as reasoning via functional groups, offers critical interpretability and verifiability advantages. Functional groups are foundational concepts in chemistry education and practice, and using them as a lens for model evaluation allows chemists to diagnose, guide, and understand a model’s capabilities. This interpretability is particularly valuable when assessing reasoning ability in a domain where structure–function relationships are subtle and complex.
>
> That said, we fully agree that one of the promises of machine learning is the potential to uncover patterns that go beyond human-designed abstractions. By establishing functional groups as a starting point for structured reasoning, FGBench does not constrain model discovery, but rather provides a controlled setting to assess whether models can go from human-guided heuristics to data-driven generalizations.
>
> > Comment #3:
> This discussion should not diminish the value of FGBench, which I think is a useful benchmark. However, I believe that the paper needs a more nuanced discussion of the definition and limitations of FGs for molecular property prediction.
>
> Response:
> We sincerely thank you for the feedback and for recognizing the value of FGBench as a benchmark. We fully agree that the paper benefits from a more nuanced discussion of the definition and limitations of functional groups (FGs) in the context of molecular property prediction.
>
> In response to this point, and as elaborated in our reply to Comment #1, we have revised the manuscript to more clearly acknowledge that FGs are a heuristic construct, whose influence on molecular properties is context-dependent, non-additive, and often intertwined with the molecular backbone and environmental conditions. We now emphasize that FGs are used in FGBench not as isolated predictors, but as reasoning anchors within a broader structural and chemical context.
>
> We hope these clarifications address your concern, and we appreciate your constructive suggestions, which have strengthened the conceptual framing of our work.
>
> > Comment #4:
> How do the authors handle changes in parts of the molecule that are not labeled as “functional groups”?
>
> Response:
> We thank the reviewer for raising this important question. In our framework, if a modified part of a molecule does not match any predefined functional group (FG) pattern, it is not discarded. Instead, we adopt the following strategy:
>
> For heteroatoms or small fragments not covered by FG definitions, we retain and describe them using their atomic identity.
>
> For carbon-based fragments, particularly linear or branched aliphatic chains, we aggregate them into alkane groups based on their length (e.g., "C4", "C5", etc.), allowing them to be treated consistently in the comparison.
>
> For more complex or ambiguous cases that cannot be reliably represented, they will be filtered out by validation by reconstruction method.
>
> This approach ensures that most structural changes, whether they involve traditional functional groups or not, are captured and represented in the dataset. It also preserves the completeness and interpretability of the QA tasks while maintaining compatibility with our reasoning framework.
>
> > Comment #5:
> “However, more fine-grained structural distinctions, such as position isomerism (e.g., ortho-, meta-, para- substitution in aromatics), carbon chain isomerism (e.g., linear vs. branched alkanes), and stereoisomerism (e.g., enantiomers and diastereomers), are not explicitly considered in this version of the dataset.” This statement is unclear. What exactly do the authors refer to? The current framework is able to represent all kinds of constitutional isomerism, and can e.g. distinguish whether a functional group is attached in the ortho, meta or para position of an arene, or whether a functional group is attached at a linear or a branched position of an alkane backbone. This statement should be revised for clarity.
>
> Response:
> Thank you for pointing out this ambiguity. You are absolutely right that the current framework can capture many forms of constitutional isomerism. The statement in the original manuscript was meant to highlight a specific limitation: our current system only captures isomeric differences when they involve functional group modifications. For example, while it can distinguish between phenol and o-cresol (due to an added methyl group), it cannot differentiate o-cresol from m-cresol, as the two differ only in the relative positions of identical functional groups, not in their presence or absence.
> Capturing such positional isomerism and subtle structural variations, where the set of FGs is unchanged but their topological arrangement varies, requires more advanced atom-level alignment, which we are actively investigating as part of future work. We will revise the relevant statement in the manuscript for clarity and add this explanation to better reflect the current scope and limitations of the dataset.

---

> > ### Comment · Reviewer_AxbV · 2025-08-03
> >
> > I thank the reviewers for their detailed responses. Most of my comments were not directed at the experiments themselves, but rather at the way the concept of functional groups is presented and discussed in the manuscript. I believe that the paper provides a useful benchmark, but the limitations and ambiguities of the functional group concept should be discussed more clearly, both in the introduction and when discussing the empirical findings.
> >
> > Let me illustrate the issues of ambiguity and context specificity: The authors refer to “tetrahydrofuran” as a functional group. From a chemists’s perspective, this is unusual. In most contexts (e.g. solubility prediction, reactivity prediction), only a part of the tetrahydrofuran scaffold would typically be considered a functional group, namely the alkoxy / ether group. There may be specialized contexts (e.g. cases of drug discovery) where treating the entire motif as a functional group could make sense. This example highlights the ambiguity in functional-group-level reasoning, which should be reflected in the paper.
> >
> > **Minor Comments**
> > > For example, while it can distinguish between phenol and o-cresol (due to an added methyl group), it cannot differentiate o-cresol from m-cresol, as the two differ only in the relative positions of identical functional groups, not in their presence or absence.
> >
> > Is this because the benzene ring is considered a functional group itself? Otherwise, the framework should be able to differentiate between the different carbon skeletons. This re-emphasizes my earlier point: The definition of functional groups is ambiguous, non-trival, and context-dependent.

---

> > > ### Author Response · Authors · 2025-08-04
> > >
> > > We sincerely thank the reviewer for the helpful clarification and illustrative examples. We fully agree with the core point that the concept of functional groups is inherently ambiguous, context-dependent, and non-trivial. The example of tetrahydrofuran is particularly insightful: in many chemical contexts, only the ether moiety would be considered the functional group, while the entire tetrahydrofuran ring may be better regarded as a scaffold or structural motif. We will incorporate a more careful discussion of these nuances in the manuscript to improve clarity and more accurately position the scope and contributions of our work.
> > >
> > > **Regarding the minor comment on o-cresol vs. m-cresol:**
> > >
> > > We appreciate the opportunity to clarify. The “validation by reconstruction” step in our pipeline does support distinguishing positional isomers such as o-cresol and m-cresol, provided that the relevant FG differences have already been correctly identified. The main limitation lies in the initial detection of FG differences between two molecules. Importantly, there is a subtle distinction between identifying FG-level differences and identifying positional isomerisms:
> > >
> > > * Identifying differences between two molecules with FG modifications typically involves three steps: detecting FG differences, removing FG differences, and then detecting residual alkyl differences.
> > >
> > > * In contrast, identifying differences between positional isomers involves: ensuring no FG differences exist, detecting positional isomerism, removing those positional differences, and finally identifying any alkyl differences.
> > >
> > > We are actively working on an extension that incorporates a routing mechanism to distinguish and handle both scenarios. We once again thank the reviewer for raising these thoughtful points, which have helped us sharpen both the conceptual framing and technical accuracy of the manuscript.

---

> > > > ### Comment · Reviewer_AxbV · 2025-08-04
> > > >
> > > > I thank the reviewers for their rapid responses. My overall positive assessment of this work remains unchanged.

---

### Official Review · Reviewer_p9wx · 2025-07-02

**Ethics Flags:** Data privacy, copyright, and consent
**Rating:** 5
**Confidence:** 3

**Summary:**

This paper introduces FGBench, a novel dataset and benchmark designed to address a critical gap in AI-driven chemistry: the inability of large language models (LLMs) to perform fine-grained reasoning about functional groups (FGs)—key substructures (e.g., -OH, -COOH) that determine molecular properties. Existing datasets (e.g., MoleculeNet) focus solely on molecule-level predictions, overlooking FG impacts. FGBench fills this void with 625K QA pairs spanning 8 molecular properties. Benchmarking six state-of-the-art LLMs (including GPT-4o and Llama-3) on a curated 7k sample subset reveals severe limitations in FG-level reasoning, particularly for multi-FG tasks (e.g., GPT-4o’s accuracy drops from 66.7% to 48.8%).

**Dataset Code Accessibility:**

Yes

**Ethical Considerations:**

Yes, there are ethics concerns that require attention by the authors

**Final Justification:**

new results have significantly strengthened the paper.

**Limitations Weaknesses:**

* The 7K benchmark subset uses at most 25 samples per task (Table 3), potentially underrepresenting rare FGs or complex interactions. Results may not generalize to infrequent cases (e.g., tetrahydrofuran in Table 1 appears in only 4.48% of data).
* Benchmarking only 6 LLMs (all general-domain except ChemLLM) overlooks specialized models (e.g., nach0, PRESTO, BioT5+).
* It's unclear how conclusions about LLM limitations may not reflect state-of-the-art chemistry AI.
* Although FGBench emphasizes interoperability by providing precise functional group annotations, the dataset's reliance on existing molecular property datasets (e.g., MoleculeNet) could limit its uniqueness.

**Strengths Contributions:**

* FGBench is the first dataset explicitly designed for functional group (FG)-level molecular property reasoning in LLMs. Prior datasets (e.g., MoleculeNet, Mol-Instructions) focus solely on molecule-level predictions, ignoring FG impacts. Explicitly contrasts with molecule-level datasets (MoleculeNet, CHEMBL) and fragment-based pretraining (FG-BERT, FARM), noting their inability to support FG-driven reasoning.
* 625K QA pairs spanning 245 FGs and 8 properties (e.g., solubility, toxicity) from 10 chemistry datasets. The dataset includes both regression and classification tasks across different functional groups.
* The results reveal that while models perform reasonably well on single functional group tasks, they struggle significantly with multiple functional group interactions, indicating a need for improved reasoning capabilities.

---

> ### Author Rebuttal · Authors · 2025-07-31
>
> We thank you for your thoughtful review and recognizing the novelty and significance of our work. Inspired by your suggestion, the new experiment reveals new insights on specialized models. We will incorporate additional experiments and discussions in the updated version of our paper. Please find our responses to your questions and concerns below.
>
> > Comment #1:
> The 7K benchmark subset uses at most 25 samples per task (Table 3), potentially underrepresenting rare FGs or complex interactions. Results may not generalize to infrequent cases (e.g., tetrahydrofuran in Table 1 appears in only 4.48% of data).
>
> Response:
> We thank the reviewer for raising this important point. We fully acknowledge that the 7K benchmark subset may underrepresent rare functional groups or complex interactions. Ideally, the benchmark would include an even distribution across all 245 functional groups and 58 tasks; however, this presents a significant challenge in practice.
>
> The benchmark subset represents a trade-off between comprehensiveness and accessibility. Our goal was to make FGBench evaluation feasible for the broader community with limited computing resources while maintaining a diverse and representative sample. Thus, we chose a task- and category-based sampling strategy to ensure balanced coverage across datasets and reasoning task types (Single, Interaction, and Comparison), rather than FG frequency.
>
> We also note that the unbalanced FG distribution is an intrinsic characteristic of chemistry datasets, where certain functional groups (e.g., hydroxyl, nitrile) appear far more frequently than others due to natural chemical and dataset biases. Some datasets (e.g., ClinTox or FreeSolv) contain only a few thousand molecules, which further limits coverage of rare FGs. We agree that future work should explore FG-aware or stratified sampling methods to more explicitly include rare or chemically significant groups in the benchmark.
>
> > Comment #2:
> Benchmarking only 6 LLMs (all general-domain except ChemLLM) overlooks specialized models (e.g., nach0, PRESTO, BioT5+).
> Response
> We thank the reviewer for the valuable suggestion regarding the inclusion of specialized models such as nach0, PRESTO, and BioT5+ in the benchmark.
>
> While the majority of the six models we evaluated are general-domain LLMs, these include widely adopted and chemistry-capable models such as GPT-4o, Qwen2.5, and LLaMA-3.1, all of which have demonstrated strong performance on chemistry tasks in recent studies [1].
>
> Our selection aimed to balance coverage and diversity within the constraints of computational resources, and spans several axes:
>
> * Open-source vs. closed-source (e.g., GPT-4o vs. LLaMA, Qwen)
>
> * Small vs. large model size (e.g., LLaMA 8B vs. 70B)
>
> * General-purpose vs. domain-specific (e.g., ChemLLM-7B)
>
> * Reasoning-optimized vs. general LLMs (e.g., o3-mini vs. others)
>
> In response to the reviewer’s suggestion, we conducted an additional evaluation of nach0, a specialized chemistry model built on the T5 architecture. We excluded BioT5+ due to its lack of support for SMILES input and PRESTO, which is primarily designed for synthetic route planning rather than property reasoning.
>
> Our analysis of nach0 revealed several limitations:
>
> 1. Lack of reasoning traceability – the model often provides direct answers without coherent or interpretable reasoning.
>
> 2. Poor instruction-following behavior – we observed a low validity rate, with many responses being irrelevant or containing nonsensical output.
>
> 3. Severe label bias – while nach0 showed superficially high accuracy on the Single Boolean category, we found that 97.7% of its valid answers (1073/1098) were “False”, indicating a strong prior bias rather than meaningful reasoning.
>
> These findings suggest that, despite strong performance on their pretraining tasks, current specialized models such as nach0 do not generalize well to new, reasoning-intensive tasks like those in FGBench. This highlights a key gap in the generalization and reasoning ability of domain-specific models, and underscores the value of FGBench in identifying these limitations.
>
> To promote broader participation and further evaluation, we have also updated the GitHub repository to include clear instructions for external contributors to submit and share their benchmark results on FGBench.
>
> |               | Single            |          |                 |          | Interaction       |          |                 |          | Comparison        |          |                 |          |
> |---------------|-------------------|----------|-----------------|----------|-------------------|----------|-----------------|----------|-------------------|----------|-----------------|----------|
> |               | Boolen (n = 1423) |          | Value (n = 375) |          | Boolen (n = 1425) |          | Value (n = 350) |          | Boolen (n = 2848) |          | Value (n = 725) |          |
> |               | ACC(↑)            | Valid(↑) | RMSE(↓)         | Valid(↑) | ACC(↑)            | Valid(↑) | RMSE(↓)         | Valid(↑) | ACC(↑)            | Valid(↑) | RMSE(↓)         | Valid(↑) |
> | GPT-4o        | 0.667             | 0.999    | 77.990          | 0.813    | 0.488             | 0.998    | 43.577          | 0.891    | 0.614             | 0.992    | 68.857          | 0.708    |
> | o3 mini       | 0.687             | 0.999    | 101.886         | 0.960    | 0.693             | 1.000    | 39.943          | 0.977    | 0.703             | 1.000    | 64.579          | 0.975    |
> | Llama-3.1 8B  | 0.548             | 0.993    | 162.351         | 0.840    | 0.547             | 0.982    | 421.325         | 0.780    | 0.474             | 0.991    | 80.566          | 0.761    |
> | Llama-3.1 70B | 0.683             | 1.000    | 84.119          | 0.973    | 0.530             | 1.000    | 38.646          | 0.977    | 0.456             | 1.000    | 64.887          | 0.943    |
> | ChemLLM-7B    | 0.233             | 0.997    | 209.584         | 0.629    | 0.235             | 0.997    | 162.742         | 0.566    | 0.25              | 1.000    | 65.428          | 0.514    |
> | Qwen2.5-7B    | 0.590             | 0.999    | 63.511          | 0.576    | 0.396             | 0.999    | 36.307          | 0.683    | 0.664             | 1.000    | 65.471          | 0.223    |
> | nach0-base    | 0.606             | 0.798    | 104.534         | 0.539    | 0.543             | 0.756    | 172.929         | 0.683    | 0.041             | 0.149    | 12221.946       | 0.879    |
>
>
> > Comment #3:
> It's unclear how conclusions about LLM limitations may not reflect state-of-the-art chemistry AI.
> Although FGBench emphasizes interoperability by providing precise functional group annotations, the dataset's reliance on existing molecular property datasets (e.g., MoleculeNet) could limit its uniqueness.
>
> Response:
> We appreciate the reviewer’s concern regarding the FGBench. We would like to emphasize that FGBench is the first benchmark to formalize and evaluate functional group–level molecular property reasoning. Prior chemistry AI efforts have largely focused on molecule-level prediction, optimization, or generation. In contrast, FGBench introduces a totally new task paradigm: assessing how specific functional group edits impact molecular properties, a reasoning task that closely mirrors how chemists think and design molecules, but has been largely absent from prior datasets and benchmarks. We believe FGBench is also a valuable dataset to develop new chemistry AI tools on it.
>
> Although FGBench utilizes existing molecular property datasets (e.g., MoleculeNet) as its source data, it introduces a novel data processing pipeline that extracts fine-grained functional group differences with precise positional information and curates 625K structured QA pairs across multiple reasoning categories (Single FG impact, Multiple FG interaction, and Molecular Comparison). This pipeline provides a new lens for interpreting molecular properties and enables new types of model evaluation, beyond what is possible in the original datasets. The pipeline is fully generalizable and can be applied to other datasets.
>
> Regarding the use of general-domain LLMs such as GPT and LLaMA, we note that such models have already demonstrated strong performance in diverse chemistry applications, including molecule optimization [1], property prediction [2], and experimental planning [3]. These tasks often require a mix of skills — understanding chemical terminology, interpreting molecular structure, and performing reasoning and planning — which are core strengths of large general-purpose LLMs. As in other domains such as mathematics, there are two complementary approaches to solving complex problems: (i) developing specialized domain-specific models (e.g., AlphaGeometry for math), and (ii) extending general LLMs with appropriate prompting and fine-tuning [4]. FGBench aims to support both directions by providing a challenging, interpretable benchmark for structure-based chemical reasoning.
>
> In this context, we believe FGBench provides a unique and valuable contribution to the chemistry AI community, not only in terms of the dataset and benchmark, but also as a foundation for developing next-generation models capable of interpretable and structure-aware chemical reasoning.
>
> [1] Yuksekgonul, Mert, et al. "Optimizing generative AI by backpropagating language model feedback." Nature 639.8055 (2025): 609-616.
>
> [2] Jablonka, Kevin Maik, et al. "Leveraging large language models for predictive chemistry." Nature Machine Intelligence 6.2 (2024): 161-169.
>
> [3] Boiko, Daniil A., et al. "Autonomous chemical research with large language models." Nature 624.7992 (2023): 570-578.
>
> [4] Huang, Yichen, and Lin F. Yang. "Gemini 2.5 Pro Capable of Winning Gold at IMO 2025." arXiv preprint arXiv:2507.15855 (2025).

---

> > ### Author Response · Authors · 2025-08-05
> > **A summary of responses**
> >
> > We greatly appreciate your time and value your insightful and comprehensive feedback. We briefly summarize our main responses based on your thoughtful comments:
> >
> > * Benchmark subset representativeness: We clarified the trade-off between diversity and accessibility in constructing the 7K benchmark subset and acknowledged the limitations due to natural FG frequency imbalances in scientific datasets.
> >
> > * Inclusion of specialized models: Inspired by your suggestion, we conducted a new evaluation of the **nach0** chemistry model. The results revealed limitations in reasoning traceability, instruction following, and generalization.
> >
> > * FGBench uniqueness and broader impact: While FGBench builds on existing property datasets, it introduces a novel functional group–centered reasoning task, a scalable data processing pipeline, and a new lens for evaluating chemical understanding in LLMs.
> >
> > We hope the new results and explanations could address your concerns. We would greatly value any further thoughts you may have, as they are important for strengthening our work.
> >
> > Thank you again for your time and attention.
> >
> > Warm regards,
> >
> > The Authors

---

> > > ### Comment · Area_Chair_J88j · 2025-08-05
> > > **Please engage with rebuttal**
> > >
> > > Dear reviewer,
> > >
> > > Thank you for your work so far reviewing this paper. The discussion period is almost over and you have yet to acknowledge and engage with the authors' rebuttal. Please respond to it, e.g., by indicating whether it addresses your concerns and questions? If it doesn't please let the authors know why, and if there are remaining questions, please ask them now to give them at fair chance at responding.
> > >
> > > Best,
> > >
> > > AC

---

> > ### Comment · Reviewer_p9wx · 2025-08-07
> >
> > > While the majority of the six models we evaluated are general-domain LLMs, these include widely adopted and chemistry-capable models such as GPT-4o, Qwen2.5, and LLaMA-3.1, all of which have demonstrated strong performance on chemistry tasks in recent studies [1].
> >
> > This claim contradicts your results from nach0, which I highlighted as a strong domain-specific language model (e.g., single accuracy: GPT-4o – 0.667, nach0-base – 0.606, LLaMA 3.1 8B – 0.548). Notably, these models differ significantly in size, with nach0 being much smaller. These results show the need for more experiments with domain-specific LLMs.
> >
> > Due to lack of domain-specific LLMs, I keep my score.

---

> > > ### Author Response · Authors · 2025-08-07
> > >
> > > We thank you for your careful observation regarding the result that the domain-specific LLM nach0 achieves a comparable accuracy on the Single Bool task to GPT-4o and LLaMA 3.1. This finding is indeed interesting, especially considering nach0’s much smaller model size, and it motivated us to further investigate the underlying cause. However, our follow-up analysis suggests that **nach0’s performance is primarily driven by output bias rather than actual model ability**, as detailed below:
> > >
> > > * **Low valid rate and limited instruction-following ability**. The valid rate and accuracy for nach0 were calculated by applying a post-processing adjustment that maps its original answers (“yes” or “no”) to the expected Boolean format (“True” or “False”). Without this adjustment, the valid rate was nearly zero. Even after adjustment, nach0 still shows a lower valid rate than most other models. We include both the original and adjusted outputs below.
> > >
> > > |                       | Single            |          |                 |          | Interaction       |          |                 |          | Comparison        |          |                 |          |
> > > |-----------------------|-------------------|----------|-----------------|----------|-------------------|----------|-----------------|----------|-------------------|----------|-----------------|----------|
> > > |                       | Bool (n = 1423) |          | Value (n = 375) |          | Bool (n = 1425) |          | Value (n = 350) |          | Bool (n = 2848) |          | Value (n = 725) |          |
> > > |                       | ACC(↑)            | Valid(↑) | RMSE(↓)         | Valid(↑) | ACC(↑)            | Valid(↑) | RMSE(↓)         | Valid(↑) | ACC(↑)            | Valid(↑) | RMSE(↓)         | Valid(↑) |
> > > | nach0-base (adjusted) | 0.606             | 0.798    | 104.534         | 0.539    | 0.543             | 0.756    | 172.929         | 0.683    | 0.041             | 0.149    | 12221.946       | 0.879    |
> > > | nach0-base (original) | 0.006             | 0.036    | 102.758         | 0.525    | 0.009             | 0.034    | 200.405         | 0.494    | 0.024             | 0.118    | 12551.381       | 0.833    |
> > >
> > >
> > > * **Output bias**. Among nach0’s valid answers on the Single Boolean task, 97.7% (1073 out of 1098) were “False.” Since the ground-truth distribution is 77.4% “False,” this bias alone leads to an accuracy of 0.606. Thus, the seemingly good performance can largely be explained by label imbalance exploitation rather than true model understanding.
> > >
> > > * **Lack of rationale**. nach0's responses consist of only “yes” or “no,” without any supporting explanation. In the case of invalid answers, the outputs were often irrelevant to the question—for example, responding with general molecular descriptions like “The molecule is a colorless, odorless powder. Non-corrosive...” when asked about property value changes.
> > >
> > > In summary, the results from nach0 do not contradict our core claim. We apologize for the ambiguity in the claim, and what we want to express is that general-domain LLMs are more capable in chemical tasks that require planning and reasoning abilities. We agree that domain-specific models such as nach0 (T5-based) or ChemLLM (pretrained and fine-tuned on chemical data) can perform well on task types they have seen during training. The FGBench poses a new challenge for them that is if they can **generalize the learned knowledge to new task types.**
> > >
> > > We believe our analysis of nach0 and ChemLLM highlights an important direction: **the need to equip domain-specific LLMs with stronger reasoning and generalization abilities**. We sincerely appreciate your comments, which prompted this deeper investigation and helped reveal the current limitations and future opportunities in domain-specific modeling for chemistry.

---

> > ### Comment · Reviewer_p9wx · 2025-08-07
> >
> > I appreciate your work on adjustment, yet my concern is lack of proper domain-specific LLMs. Looks like there is no need for such models since GPT-4o gave 0.999 valid results according to your table.
> >
> > > We agree that domain-specific models such as nach0 (T5-based) or ChemLLM (pretrained and fine-tuned on chemical data) can perform well on task types they have seen during training. The FGBench poses a new challenge for them that is if they can generalize the learned knowledge to new task types.
> >
> > Yes, this could be true since nach0 is 300M param. model. What about models trained on MolInstructions? Is MolInstructions the largest chem dataset with NL-SMILES/SELFIE pairs up to date?

---

> > > ### Author Response · Authors · 2025-08-08
> > >
> > > > I appreciate your work on adjustment, yet my concern is lack of proper domain-specific LLMs.
> > >
> > > We thank you for your comment. Your valuable concern is not just a concern, but the current situation that lack of domain-specific (chemical) LLMs with reasoning and generalization capabilities. None of the domain-specific LLMs we tested (ChemLLM, nach0, and Llama-3-8B-MolInst) is able to produce long context or reasoning process to elaborate their answer.
> > >
> > > > Looks like there is no need for such models since GPT-4o gave 0.999 valid results according to your table.
> > >
> > > GPT-4o gave 0.999 valid results mean it can understand the question and the requirement of the question, but the accuracy of GPT-4o indicate it cannot understand chemistry very well. **The primary purpose of our dataset and benchmark is to highlight this gap and emphasize the need to develop domain-specific LLMs with improved generalizability, reasoning ability, and structure understanding.**
> > >
> > > The benchamrk results also indicate that building domain-specific (chemical) LLMs with T5 architecture (nach0) or instruction finetuning (ChemLLM, Llama-3-8B-MolInst) is not enough to allow model generalize the chemical knowledge to broader tasks. FGBench challenges domain-specific (chemical) LLMs on how to incorporate them with structure understanding. We believe FGBench will inspire the development of domain specific models with structural modalities or reinforcement learning based on structure fact.
> > >
> > > >  What about models trained on MolInstructions? Is MolInstructions the largest chem dataset with NL-SMILES/SELFIE pairs up to date?
> > >
> > > We thank you for your question. MolInstructions is the largest dataset with SELFIE pairs cover both chemistry and biochemistry. We list the result of Llama3-8B model finetuned by MolInstructions on FGBench:
> > >
> > > |                               | Single            |          |                 |          | Interaction       |          |                 |          | Comparison        |          |                 |          |
> > > |-------------------------------|-------------------|----------|-----------------|----------|-------------------|----------|-----------------|----------|-------------------|----------|-----------------|----------|
> > > |                               | Boolen (n = 1423) |          | Value (n = 375) |          | Boolen (n = 1425) |          | Value (n = 350) |          | Boolen (n = 2848) |          | Value (n = 725) |          |
> > > |                               | ACC(↑)            | Valid(↑) | RMSE(↓)         | Valid(↑) | ACC(↑)            | Valid(↑) | RMSE(↓)         | Valid(↑) | ACC(↑)            | Valid(↑) | RMSE(↓)         | Valid(↑) |
> > > | Llama-3-8B-MolInst (original) | 0.107             | 0.203    | 0.388           | 0.003    | 0.059             | 0.144    | 0.000           | 0.000    | 0.469             | 0.873    | 20.871          | 0.001    |
> > > | Llama-3-8B-MolInst (adjusted) | 0.107             | 0.203    | 328.935         | 0.496    | 0.059             | 0.149    | 188.376         | 0.486    | 0.469             | 0.873    | 138.654         | 0.837    |
> > >
> > > In Comparison category, it gets comparable results as Llama-3.1-8B, while it get worse performance for others.
> > >
> > > **Some additional comments:** The motivation of design FGBench is not to simply build a instruction QA set that be used to do instruction fine-tuning and get high accuracy and low RMSE. The behavior we expect to see in domain specific LLMs is: (1) when the model knows that a property of ethane (CC) is 0.5 and the property of ethanol (CCO) is 0.6. When we ask the model what is the property of ethane after adding a hydroxyl group (CC + OH), the model can infer it is ethanol and tell the property. (2) When we ask the model how is the property change when adding a hydroxyl group to propane (CCC + OH), the model can based on the knowledge of CC vs. CCO to infer it is increase.

---

> > > > ### Comment · Reviewer_p9wx · 2025-08-08
> > > >
> > > > Thank you for the detailed answer — this has been a very constructive conversation. I am now leaning more positively toward raising the final score.
> > > >
> > > > >Your valuable concern is not just a concern, but the current situation that lack of domain-specific (chemical) LLMs with reasoning and generalization capabilities. None of the domain-specific LLMs we tested (ChemLLM, nach0, and Llama-3-8B-MolInst) is able to produce long context or reasoning process to elaborate their answer.
> > > >
> > > > 1) Do you believe that a large-scale chemistry-specific LLM (non-multimodal) is the key direction for future research? For example, I previously mentioned the PRESTO model (EMNLP paper) with graph alignment, which outperforms some existing LMs.
> > > >
> > > > 2) I recall that several chemistry datasets were published at this conference last year. Could you summarize whether these datasets could meaningfully contribute to building a large chemical foundation model? I am very curious.

---

> > ### Author Response · Authors · 2025-08-09
> >
> > We sincerely appreciate your comments, which have inspired us with many valuable new insights.
> >
> > > Do you believe that a large-scale chemistry-specific LLM (non-multimodal) is the key direction for future research? For example, I previously mentioned the PRESTO model (EMNLP paper) with graph alignment, which outperforms some existing LMs.
> >
> > Thank you for this insightful question. Both non-multimodal and multimodal chemistry-specific LLMs represent promising and open-ended research directions. At this stage, we cannot assert which will dominate. However, we expect both to continue developing in parallel over the near future.
> >
> > * Chemistry-specific LLMs (non-multimodal): the direction would be how to let the model understand overall molecular geometry and substructure information from SMILES/SELFIES; whether to use canonical SMILES or leverage the diversity of SMILES (multi SMILES can represent a single molecule), akin to a diverse pretraining corpus for enhancing the generalizability of LLMs.
> >
> > * Chemistry-specific multimodal text+molecule LLMs: the direction would be choices of molecule tokenization strategies (atom-level vs. fragment-level, GNN-based or diffusion models), and whether similar approaches from ViT or VQ-VAE can be adapted. Improving training efficiency and scalability to larger model sizes are also major topics.
> >
> > * For both categories: there is significant potential in developing chemistry-specific reinforcement learning to enhance reasoning ability, as well as designing new fine-tuning, alignment, or pretraining strategies tailored to molecular and chemistry-related tasks.
> >
> > > I recall that several chemistry datasets were published at this conference last year. Could you summarize whether these datasets could meaningfully contribute to building a large chemical foundation model?
> >
> > Certainly. Below are notable datasets and their potential contributions:
> >
> > Published at last year’s conference:
> >
> > * MolPuzzle: Infers molecular structure from diverse spectral data.
> > * MassSpecGym: Identifies molecules from MS/MS spectra.
> >
> > Other recent datasets (not from this conference):
> > * ChemData (7M): Instruction-tuning dataset compiled from PubChem, ChEMBL, ChEBI, ZINC, USPTO, ...
> > * SMolInstruct (3M): Instruction-tuning dataset integrating PubChem, MoleculeNet, Mol-Instructions, USPTO, ...
> > * Mol-Instructions (2M): Instruction-tuning dataset (molecule + protein) including PubChem, USPTO, UniProtKB, LOTUS, ...
> >
> > Compared to general-domain corpora, chemical corpora are relatively small. We believe all of the datasets mentioned are valuable, and the key challenge lies in how to fully utilize them. As illustrated by the ethane and ethanol examples discussed in the last comment, even a small number of examples can yield valuable and transferable knowledge. FGBench makes an initial attempt on molecular property, and we anticipate that the underlying logic and methodology behind FGBench could be further applied to other molecular tasks.
> >
> > In addition, we added the result of LlaSMol-Mistral-7B that based on mistralai/Mistral-7B-v0.1 and tuned on SMolInstruct dataset. The performance is slightly better than Llama-3-8B-MolInst, which might contribute to LlaSMol uses SMILES to represent molecules that is consistent with FGBench.
> >
> > |                               | Single            |          |                 |          | Interaction       |          |                 |          | Comparison        |          |                 |          |
> > |-------------------------------|-------------------|----------|-----------------|----------|-------------------|----------|-----------------|----------|-------------------|----------|-----------------|----------|
> > |                               | Boolen (n = 1423) |          | Value (n = 375) |          | Boolen (n = 1425) |          | Value (n = 350) |          | Boolen (n = 2848) |          | Value (n = 725) |          |
> > |                               | ACC(↑)            | Valid(↑) | RMSE(↓)         | Valid(↑) | ACC(↑)            | Valid(↑) | RMSE(↓)         | Valid(↑) | ACC(↑)            | Valid(↑) | RMSE(↓)         | Valid(↑) |
> > | LlaSMol-Mistral-7B (adjusted) | 0.387             | 0.922    | 266.720         | 0.923    | 0.298             | 0.968    | 262.550         | 0.983    | 0.239             | 1.000    | 245.298         | 0.924    |

---

> > > ### Author Response · Authors · 2025-08-09
> > > **Brief summary of new experiment results**
> > >
> > > We sincerely appreciate your constructive comments and questions. As the discussion period comes to a close, we have summarized the new results during the discussion in the table below. We have added additional three domain-specific models, including nach0, Llama-3-8B-MolInst, and LlaSMol-Mistral-7B. Along with ChemLLM, these models collectively cover three large fine-tuning datasets (ChemData, SMolInstruct, and Mol-Instructions) and four different architectures: InternLM2, T5, Llama, and Mistral.
> > >
> > > |                               | Single            |          |                 |          | Interaction       |          |                 |          | Comparison        |          |                 |          |
> > > |-------------------------------|-------------------|----------|-----------------|----------|-------------------|----------|-----------------|----------|-------------------|----------|-----------------|----------|
> > > |                               | Boolen (n = 1423) |          | Value (n = 375) |          | Boolen (n = 1425) |          | Value (n = 350) |          | Boolen (n = 2848) |          | Value (n = 725) |          |
> > > |                               | ACC(↑)            | Valid(↑) | RMSE(↓)         | Valid(↑) | ACC(↑)            | Valid(↑) | RMSE(↓)         | Valid(↑) | ACC(↑)            | Valid(↑) | RMSE(↓)         | Valid(↑) |
> > > | ChemLLM-7B                    | 0.233             | 0.997    | 209.584         | 0.629    | 0.235             | 0.997    | 162.742         | 0.566    | 0.25              | 1.000    | 65.428          | 0.514    |
> > > | **nach0-base** (adjusted)         | 0.606             | 0.798    | 104.534         | 0.539    | 0.543             | 0.756    | 172.929         | 0.683    | 0.041             | 0.149    | 12221.946       | 0.879    |
> > > | **Llama-3-8B-MolInst** (adjusted) | 0.107             | 0.203    | 328.935         | 0.496    | 0.059             | 0.149    | 188.376         | 0.486    | 0.469             | 0.873    | 138.654         | 0.837    |
> > > | **LlaSMol-Mistral-7B** (adjusted) | 0.387             | 0.922    | 266.720         | 0.923    | 0.298             | 0.968    | 262.550         | 0.983    | 0.239             | 1.000    | 245.298         | 0.924    |

---

> > > > ### Comment · Reviewer_p9wx · 2025-08-09
> > > >
> > > > Thank you for providing the additional results and clarifications. The new results have significantly strengthened the paper. I am happy to increase my score.

---

### Official Review · Reviewer_68Pq · 2025-07-04

**Rating:** 5
**Confidence:** 2

**Summary:**

This paper addresses the critical gap in current large language models (LLMs) for molecular property prediction by introducing FGBench, the first multi-source benchmark dataset specifically designed for functional group (FG)-level molecular property reasoning. Comprising 625,000 molecular property reasoning problems across 245 functional groups and 27 alkane chain lengths, the dataset is curated from 10 existing databases (e.g., MoleculeNet, including ESOL, HIV, and QM9). Leveraging an innovative validation-by-reconstruction strategy and the AccFG tool, it provides precise annotations of FG differences and positional information. The benchmark features three task dimensions (single-FG impact, multi-FG interactions, and direct molecular comparisons) with both Boolean and value-based QA formats.

**Dataset Code Accessibility:**

Yes

**Ethical Considerations:**

No, there are no or only very minor ethics concerns

**Final Justification:**

While the proposed dataset has limitations, notably its reliance on single-value measurements and exclusively pairwise functional group modifications, it establishes an important benchmark through its incorporation of functional group heuristics, which could potentially advance future research in domain-specific AI reasoning for functional group-level molecular property prediction.

**Limitations Weaknesses:**

The dataset focuses primarily on simple functional group modifications while lacking representation of complex structural features like stereochemistry, polycyclic systems, and organometallic compounds that are crucial in real-world drug discovery.
Molecular properties are treated as static, single-value measurements without accounting for experimental variability, condition-dependence (e.g., pH/temperature effects), or the full spectrum of extreme values seen in practice.
Multi-functional group effects are only captured through pairwise modifications, potentially missing higher-order synergistic interactions and complex structure-activity relationships.
All predictions are made without considering critical real-world factors like protein binding environments or solvent effects that significantly influence molecular properties in practical applications.

**Strengths Contributions:**

By employing Tanimoto similarity filtering (threshold >0.7) and atom-level difference validation, the dataset ensures precise annotation of 42,967 molecular comparison pairs with verified functional group modifications.
It covers 8 molecular properties (e.g., solubility, inhibitory activity) and three reasoning scenarios for holistic model evaluation.
It provides precise FG localization and SMILES-based structural annotations, which enable potential integration with graph-based or 3D molecular representation methods.
Full dataset (on HuggingFace) and templated QA generation methods are provided for extensibility.

---

> ### Author Rebuttal · Authors · 2025-07-31
>
> We thank your for your positive feedback and the recognition of the precision, integration potential, and extensibility of the dataset and the benchmark. Please find our responses to your questions and concerns below, which we will also implement together within the updated version of the paper.
>
> > Comment #1:
> The dataset focuses primarily on simple functional group modifications while lacking representation of complex structural features like stereochemistry, polycyclic systems, and organometallic compounds that are crucial in real-world drug discovery.
>
>
> Response:
> We appreciate the reviewer’s thoughtful comment regarding the scope of structural features represented in FGBench. While it is true that stereochemistry is not explicitly included in this version of the dataset, we would like to clarify that FGBench does handle complex structures, including polycyclic compounds and a subset of organometallic compounds. These are handled through our general framework without any special exclusion (see examples below).
>
> Regarding stereochemistry, we fully agree that it is an important and nuanced factor in real-world drug discovery and structure–activity relationship (SAR) analysis. However, our current focus is on functional group (FG)-level reasoning, which provides a more systematic and well-characterized foundation for evaluating model performance. Unlike FGs, even expert chemists often struggle to reason about stereochemistry’s effects.
>
> The FGBench pipeline is designed with extensibility, and it's a standardized framework that could support future incorporation of stereochemical annotations. Extending the dataset to handle atom-level variations such as R/S stereoisomers and E/Z isomerism is a high-priority direction in our future work.
>
> * Example on polycyclic systems:
>
>     * Cholesterol: `[CH3:0][CH:1]([CH3:2])[CH2:3][CH2:4][CH2:5][C@@H:6]([CH3:7])[C@H:8]1[CH2:9][CH2:10][C@H:11]2[C@@H:12]3[CH2:13][CH:14]=[C:15]4[CH2:16][C@@H:17]([OH:18])[CH2:19][CH2:20][C@:21]4([CH3:22])[C@H:23]3[CH2:24][CH2:25][C@:26]12[CH3:27]`
>         * FGs: `[('Secondary hydroxyl', 1, [(18,)])]`
>
>     * Cholesterol sulfate: `[CH3:0][CH:1]([CH3:2])[CH2:3][CH2:4][CH2:5][C@@H:6]([CH3:7])[C@H:8]1[CH2:9][CH2:10][C@H:11]2[C@@H:12]3[CH2:13][CH:14]=[C:15]4[CH2:16][C@@H:17]([O:18][S:19](=[O:20])(=[O:21])[OH:22])[CH2:23][CH2:24][C@:25]4([CH3:26])[C@H:27]3[CH2:28][CH2:29][C@:30]12[CH3:31]`
>         * FGs: `[('Sulfuric monoester', 1, [(19, 20, 21, 22, 18)])]`
>
> * Example on organometallic compounds:
>     * Ferrocene: `[Fe+2:0].[cH:1]1[cH:2][cH:3][cH-:4][cH:5]1.[cH:6]1[cH:7][cH:8][cH-:9][cH:10]1`
>     * 1,1'-Bis(diphenylphosphino)ferrocene: `[Fe+2:0].[cH:19]1[cH:20][cH:21][c:22]([P:23]([c:24]2[cH:25][cH:26][cH:27][cH:28][cH:29]2)[c:30]2[cH:31][cH:32][cH:33][cH-:34]2)[cH:35][cH:36]1.[cH:1]1[cH:2][cH:3][c:4]([P:5]([c:6]2[cH:7][cH:8][cH:9][cH:10][cH:11]2)[c:12]2[cH:13][cH:14][cH:15][cH-:16]2)[cH:17][cH:18]1`
>         * FGs: `[('Phosphine', 2, [(5,), (23,)]), ('Benzene', 4, [(1, 2, 3, 4, 17, 18), (6, 7, 8, 9, 10, 11), (19, 20, 21, 22, 35, 36), (24, 25, 26, 27, 28, 29)])]`
>
> > Comment #2:
> Molecular properties are treated as static, single-value measurements without accounting for experimental variability, condition-dependence (e.g., pH/temperature effects), or the full spectrum of extreme values seen in practice. Multi-functional group effects are only captured through pairwise modifications, potentially missing higher-order synergistic interactions and complex structure-activity relationships.
>
> Response:
> We thank the reviewer for raising these two important concerns. Regarding experimental variability and condition dependence, we agree that molecular properties can be strongly influenced by factors such as pH, temperature, and solvent conditions, and that incorporating these variables would enable richer and more realistic modeling. However, such condition-specific annotations are rarely available in existing datasets at scale. To minimize variability, we carefully selected datasets where measurements were performed under standardized and well-documented conditions. For example, the ESOL dataset reports water solubility measured at 25 °C, and the Lipophilicity dataset reports logD values measured at pH 7.4. By relying on such standardized datasets, we aim to control for extraneous variables and focus our analysis on the structural factors, particularly functional group differences, that influence properties. We fully agree that exploring context-aware property reasoning under varying experimental conditions is an important and promising future direction
>
> Regarding multi-functional group effects, we recognize that our benchmark focuses on pairwise functional group modifications. However, we would like to emphasize that each input QA pair includes the entire molecular scaffold along with the set of added/removed FGs, allowing models to potentially capture higher-order interactions beyond just pairwise effects. While we do not explicitly enumerate all possible FG combinations, the presence of multiple co-occurring groups within the molecular context allows for the emergence of complex structure–activity relationships.
>
> > Comment #3
> All predictions are made without considering critical real-world factors like protein binding environments or solvent effects that significantly influence molecular properties in practical applications.
>
> Response:
> We thank the reviewer for highlighting the important consideration of real-world biological and environmental factors, such as protein binding environments and solvent effects, in molecular property prediction.
>
> We would like to clarify that some of the molecular properties in FGBench are indeed measured under specific environmental contexts. For example, lipophilicity (logD) values are defined based on octanol/water partitioning at pH 7.4, and solvation free energy (FreeSolv) reflects aqueous solvation behavior. These standardized solvent conditions are inherent to the definitions of the properties and are preserved in our dataset to provide consistent context for functional group reasoning. As such, solvent effects are partially accounted for in these tasks.
>
> Regarding protein binding environments, we fully agree that such context plays a critical role in real-world pharmacological applications. However, for several tasks included in FGBench (e.g., HIV, Tox21, SIDER), the exact protein targets are unknown or ambiguous, and even for tasks with defined targets (e.g., BACE), the presence of multiple or non-specific binding sites complicates their inclusion. Incorporating protein context would require well-characterized protein–ligand complexes, which are not uniformly available across datasets.
>
> Nonetheless, we agree that integrating protein sequence or structure information is a promising direction for future work. We are currently exploring ways to incorporate target-aware reasoning, such as by including protein sequences or embeddings. However, it is important to note that current LLMs exhibit limited capability in interpreting protein sequences, especially in the context of fine-grained structure-activity reasoning. We will add this discussion to the manuscript to inform readers about the current scope of FGBench and the future directions.

---

> > ### Comment · Area_Chair_J88j · 2025-08-05
> > **Please engage with rebuttal**
> >
> > Dear reviewer,
> >
> > Thank you for your work so far reviewing this paper. The discussion period is almost over and you have yet to acknowledge and engage with the authors' rebuttal. Please respond to it, e.g., by indicating whether it addresses your concerns and questions? If it doesn't please let the authors know why, and if there are remaining questions, please ask them now to give them at fair chance at responding.
> >
> > Best,
> >
> > AC

---

### Official Review · Reviewer_9uzs · 2025-07-19

**Rating:** 4
**Confidence:** 3

**Summary:**

This paper introduces FGBench, a new large-scale dataset and benchmark designed to evaluate the molecular property reasoning capabilities of Large Language Models (LLMs) at the functional group (FG) level. The authors argue that existing benchmarks primarily focus on whole-molecule property prediction, neglecting the fine-grained reasoning that is crucial for chemical understanding and tasks like drug discovery. FGBench consists of 625,000 question-answer (QA) pairs derived from public datasets, covering 245 functional groups. The QA pairs are structured into three reasoning categories: (1) single functional group impacts, (2) multiple functional group interactions, and (3) direct molecular comparisons. To construct this dataset, the authors developed a novel data processing pipeline featuring a "validation-by-reconstruction" strategy to ensure high-quality molecular comparisons with precisely annotated FG differences. Benchmarking several state-of-the-art LLMs on a 7,000-sample subset of FGBench reveals that current models struggle with FG-level reasoning, particularly when multiple FGs interact. The paper makes a strong case for the need to develop more structure-aware and interpretable LLMs for chemistry.

**Dataset Code Accessibility:**

Yes

**Ethical Considerations:**

No, there are no or only very minor ethics concerns

**Final Justification:**

Thanks for the responses. After reading the responses, my concerns have been addressed. I would think this bench will make contributions to the related community, and voted for borderline accpet.

**Limitations Weaknesses:**

- The most significant limitation, as acknowledged by the authors, is that the current version of FGBench does not explicitly account for several important types of isomerism, such as positional isomerism (e.g., ortho-, meta-, para- substitution) and stereoisomerism (e.g., enantiomers). These structural nuances can have profound effects on molecular properties, and their absence limits the depth of chemical reasoning that can be evaluated. Incorporating these would be a valuable next step.

- The dataset construction process relies on a Tanimoto similarity threshold (>0.7) to pre-filter molecular pairs for computational efficiency. While practical, this may exclude structurally diverse pairs that still differ by only a few functional groups, potentially biasing the dataset towards more structurally similar comparisons.

- The authors note that for the "Molecular Comparison" task, models might achieve better performance simply due to having seen similar molecules from the underlying MoleculeNet dataset during pre-training, rather than through genuine reasoning. This makes it slightly more difficult to interpret the results for this specific task category as a pure measure of reasoning ability.

**Strengths Contributions:**

- The work moves beyond simple molecular property prediction to a more nuanced and practical task: reasoning about property changes based on specific structural modifications at the functional group level. This aligns more closely with the reasoning processes of human chemists in structure-activity relationship analysis and molecular design.

- The introduced FGBench, with 625K QA pairs,  is well-structured, with three distinct reasoning dimensions and two types of questions (Boolean and value-based) that allow for a multifaceted evaluation of LLM capabilities. The inclusion of precise positional data for FGs is a critical detail that enhances the dataset's utility.

- The proposed "validation-by-reconstruction" strategy sounds general and applicable.

- The paper provides a comprehensive benchmark of six prominent closed-source and open-source LLMs. The failure analysis further provides insights of how current models fail.

---

> ### Author Rebuttal · Authors · 2025-07-31
>
> Thank you for the thoughtful and encouraging feedback. We appreciate your recognition of our contributions, including the functional group-level reasoning task, the generalizable validation strategy, and the comprehensive benchmarking and analysis. Please find our responses to your questions and concerns below, which we will also implement together within the updated version of the paper.
>
> > Comment #1:
> The most significant limitation, as acknowledged by the authors, is that the current version of FGBench does not explicitly account for several important types of isomerism, such as positional isomerism (e.g., ortho-, meta-, para- substitution) and stereoisomerism (e.g., enantiomers). These structural nuances can have profound effects on molecular properties, and their absence limits the depth of chemical reasoning that can be evaluated. Incorporating these would be a valuable next step.
>
>
> Response:
> We sincerely thank the reviewer for highlighting this important point and for acknowledging the limitation we discussed in our manuscript. As correctly noted, the current version of FGBench focuses on reasoning at the functional group (FG) level and does not explicitly incorporate finer-grained structural distinctions such as positional isomerism or stereoisomerism. Our decision to center this initial version of FGBench on FG-level annotations was motivated by the following considerations:
>
> 1. Functional groups represent a foundational and widely understood unit of chemical reasoning in both chemical education and research.
>
> 2. FG-level effects on molecular properties are well-documented in literature and textbooks, making them an ideal starting point for benchmarking and interpreting LLMs' chemical reasoning abilities.
>
> We fully agree that more fine-grained atom-level distinctions including positional and stereoisomeric variations play a critical role in molecular properties. Incorporating these into future versions of FGBench is an important extension of our current work.
>
> >Comment #2:
> The dataset construction process relies on a Tanimoto similarity threshold (>0.7) to pre-filter molecular pairs for computational efficiency. While practical, this may exclude structurally diverse pairs that still differ by only a few functional groups, potentially biasing the dataset towards more structurally similar comparisons.
>
> Response:
> We thank the reviewer for this thoughtful observation regarding the use of a Tanimoto similarity threshold (>0.7) in our dataset construction pipeline. Indeed, this filtering step prioritizes structurally similar molecular pairs and may exclude some structurally diverse examples that nonetheless differ by only a few functional groups.
>
> Our choice of this threshold was guided by both computational practicality and chemical reasoning considerations. In particular, chemists often analyze functional group effects by varying specific substituents on a fixed molecular scaffold. This is a principle seen in substrate scope analysis [1] and drug screening [2]. In these contexts, holding the scaffold constant allows for more controlled and interpretable comparisons of how specific FGs influence molecular properties.
>
> [1] Chen, Bin, et al. "Steering oxygen-centred radicals with ground-state ene-reductases for enantioselective intermolecular hydroalkoxylations." Nature Catalysis (2025): 1-9.
> [2] Balint, Gabor A. "Artemisinin and its derivatives: an important new class of antimalarial agents." Pharmacology & therapeutics 90.2-3 (2001): 261-265.
>
> > Comment #3:
> The authors note that for the "Molecular Comparison" task, models might achieve better performance simply due to having seen similar molecules from the underlying MoleculeNet dataset during pre-training, rather than through genuine reasoning. This makes it slightly more difficult to interpret the results for this specific task category as a pure measure of reasoning ability.
>
> Response:
> We thank the reviewer for this insightful comment. We agree that the "Molecular Comparison" task category may be influenced by the fact that some models could achieve higher performance by memorizing molecular-level patterns rather than engaging in genuine functional group-level reasoning. However, the role of "Molecular Comparison" is still important. We would like to clarify that the Molecular Comparison task serves two key purposes:
>
> 1. It is a reasoning task, where models must compare two molecules and infer relative property differences without explicit FG guidance. It is a more complex task than the other two tasks as it requires an additional step to tell the difference between two molecules.
> 2. It also functions as a reference baseline, allowing us to contrast model performance with the Single and Interaction tasks. This comparison helps assess whether a model's performance is driven by reasoning over functional group changes or merely memorization of whole-molecule patterns. For instance, if a model performs better on Single and Interaction tasks than on Molecular Comparison, it suggests that the model has developed some genuine functional group-level reasoning ability. Conversely, if its performance is disproportionately better on the Comparison task, it may indicate reliance on molecule-level memorization or dataset overlap.

---

> > ### Comment · Area_Chair_J88j · 2025-08-05
> > **Please engage with rebuttal**
> >
> > Dear reviewer,
> >
> > Thank you for your work so far reviewing this paper. The discussion period is almost over and you have yet to acknowledge and engage with the authors' rebuttal. Please respond to it, e.g., by indicating whether it addresses your concerns and questions? If it doesn't please let the authors know why, and if there are remaining questions, please ask them now to give them at fair chance at responding.
> >
> > Best,
> >
> > AC

---

### Note · Authors · 2025-08-14

We sincerely thank the reviewers and ACs for taking the time to review our paper! We are delighted that our contributions have been recognized by all reviewers. FGBench is a dataset and benchmark for molecular property reasoning in large language models (LLMs) with functional group-level annotations. It aims to advance the development of structure-aware LLMs with stronger reasoning capabilities in the chemistry domain.

During the rebuttal period, we engaged in thorough and constructive discussions with the reviewers. These discussions not only addressed the reviewers’ concerns but also inspired many new insights for future extensions of this work and developing new chemistry-specific LLMs.

1. **Accounting for isomerism** (9uzs, 68Pq, AxbV): The current scope of this work focuses on functional groups. We further elaborated on the generalizability of our approach, including the capability of “validation by reconstruction” pipeline to handle isomerism, and outlined a feasible technical route to capture isomeric differences in the future  work (see the response to AxbV).

2. **Task selection** (9uzs, 68Pq, p9wx): The tasks in FGBench cover common molecular properties but can be readily extended to other datasets, including those involving polycyclic systems and organometallic compounds (see the response to 68Pq). Furthermore, the chosen properties and designed task categories reveal a key limitation: the limited structure awareness and reasoning ability of current LLMs and domain-specific LLMs (see the response to p9wx).

3. **Clarifying ambiguity in functional groups** (AxbV): We will clarify in the manuscript that the definition of functional groups can be context-dependent and non-trivial, to help readers better understand the scope and interpretation of functional groups in this work (see the response to AxbV).

Thank you for your attention.

Best regards,

The Authors

---

### Decision · Program_Chairs · 2025-09-18

**Decision:**

Accept (poster)

**Comment:**

This paper introduces FGBench, a large, open dataset and benchmark targeting functional-group (FG)–level molecular property reasoning with LLMs. The dataset comprises ~625K QA pairs spanning 245 functional groups and 8 molecular properties, built from ~10 established sources (e.g., MoleculeNet). Questions cover three reasoning categories: (1) single-FG impact, (2) multi-FG interactions, and (3) direct molecular comparisons, with both Boolean and value (regression) targets. A “*validation-by-reconstruction*” pipeline pairs structurally similar molecules and localizes FG edits (including positions) to produce high-precision comparisons.

The benchmark evaluates several general-purpose and chemistry-tuned LLMs on a ~7K curated test split. A consistent empirical pattern emerges: models do reasonably well on single-FG questions but degrade substantially on multi-FG interactions and some comparison tasks, indicating that current LLMs struggle with context-dependent, non-additive FG reasoning.

Compliance: Code, data (Hugging Face), and metadata are available; Croissant validation passes

### Strengths
- Clear DB contribution & scope. FGBench is a dataset & benchmark focused on evaluation, not method novelty, hence well aligned with the track. It introduces a new granularity (FG-level) that is scientifically meaningful and under-served by prior molecule-level datasets.
- Careful curation & technical soundness. The validation-by-reconstruction pipeline and explicit FG localization (including positional indices) are compelling; this makes the resource valuable for structure-aware methods (text+SMILES/graphs/3D).
- Insightful empirical takeaway. Across multiple strong LLMs, the multi-FG interaction gap is robust, a finding that is likely to prove useful for future model design and standardized evaluation.
- Accessibility & reproducibility. Public dataset (HF), code (GitHub), Croissant metadata, and clear loading instructions; ethics review found no major concerns beyond a licensing acknowledgement, which authors committed to add.
- Community impact. The paper motivates interpretable, chemistry-anchored reasoning and creates a common testbed likely to catalyze follow-up work (e.g., multimodal or tool-augmented models).

### Weaknesses
- Representativeness of the 7K test split (Reviewer p9wx, partially addressed in rebuttal). The curated benchmark subset uses at most ~25 samples per task and may underrepresent rare functional groups or complex interactions. This raises questions about how well benchmark results generalize to infrequent cases.
- Scope regarding isomerism and context (Reviewers 9uzs and AxbV). The current dataset does not comprehensively capture positional or stereoisomerism, which can strongly affect molecular properties. In addition, AxbV emphasized that functional groups are heuristic, ambiguous, and context-dependent, and that the manuscript should explicitly discuss this limitation.
- Pre-filtering bias (Reviewer 9uzs). The reliance on a Tanimoto similarity threshold (>0.7) may bias the dataset toward scaffold-similar pairs and exclude structurally diverse but chemically relevant comparisons.
- Limited structural and contextual coverage (Reviewer 68Pq). The dataset focuses primarily on functional group edits and does not incorporate stereochemistry, polycyclic systems, or organometallic chemistry in a systematic way. It also treats properties as static single values, omitting condition-dependence (pH, solvent, protein binding), and restricts interactions to pairwise FG changes, potentially missing higher-order effects.
- Uncertainty reporting (general). Model performance is reported mainly as single-point accuracy or RMSE; more variance estimates, confidence intervals, or prompt sensitivity analyses would strengthen the reliability of the benchmark.
- Baseline breadth (Reviewer p9wx, partially addressed in rebuttal). The initial evaluation emphasized general-purpose LLMs with only one chemistry-specific baseline (ChemLLM). Although the rebuttal added nach0, Llama-3-8B-MolInst, and LlaSMol-Mistral-7B, broader exploration of structure-aware or multimodal baselines (text + graph/3D) is still missing.

### Decision Rationale
- **High-quality resource squarely in D&B scope.** The dataset/benchmark is substantial, thoughtfully engineered, openly released, and easy to use, therefore a great fit for the D&B track.
- **Actionable scientific insight.** The consistent multi-FG reasoning deficit across models is a clear and valuable finding for the community.
- **Responsible handling of limitations.** Reviewers raised salient points on FG ambiguity, isomerism, and sampling; the authors acknowledged these and clarified scope, and added experiments that improved the paper’s evidentiary base.

### Discussion and Rebuttal Summary

Reviewer p9wx
- Concerns:
  - Small per-task sample sizes in the 7K benchmark subset, potentially underrepresenting rare functional groups.
  - Limited coverage of domain-specific chemistry LLMs; unclear whether conclusions reflect state-of-the-art chemistry AI.
  - Questioned the uniqueness of FGBench given reliance on MoleculeNet.
- Author response:
  - Clarified sampling trade-offs and dataset biases.
  - Added experiments with nach0, Llama-3-8B-MolInst, and LlaSMol-Mistral-7B; analyzed nach0’s apparent performance as label imbalance bias with low validity.
  - Emphasized FGBench’s novelty as a functional group–level reasoning dataset with a new processing pipeline.
- Effect: Concerns were largely resolved. Reviewer acknowledged that the new results significantly strengthened the paper and raised their score.

Reviewer 9uzs
- Concerns:
  - Lack of explicit treatment of isomerism (positional, stereo).
  - Bias introduced by Tanimoto >0.7 filtering.
  - Difficulty in interpreting results of the Molecular Comparison task, since models may rely on dataset overlap rather than reasoning.
- Author response:
  - Acknowledged isomerism as future work, while clarifying the scope of FG-level reasoning.
  - Justified pre-filtering as chemically reasonable (aligns with scaffold-focused analysis).
  - Explained that the Comparison task serves as both a reasoning challenge and a reference baseline.
- Effect: Reviewer stated their concerns were addressed and maintained a borderline accept recommendation.

Reviewer 68Pq
- Concerns:
  - Limited coverage of stereochemistry, polycyclic systems, organometallics.
  - Properties treated as static single values without variability or condition dependence.
  - Multi-FG effects restricted to pairwise interactions.
  - Lack of protein/solvent context in tasks.
- Author response:
  - Clarified that polycyclics and some organometallics are included; stereochemistry remains for future work.
  - Pointed out standardized experimental conditions in source datasets (e.g., pH 7.4 for Lipophilicity).
  - Explained that pairwise FG modifications still allow complex patterns to emerge in context.
  - Acknowledged protein binding as an important but currently impractical extension.
- Effect: Reviewer accepted these as reasonable scope limitations and retained their positive evaluation.

Reviewer AxbV
- Concerns:
  - Ambiguity and context-dependence of functional groups; risk of overstating FG influence.
  - Need for a nuanced discussion of FG heuristics and limitations.
  - Clarifications requested on non-FG edits and positional isomer handling.
- Author response:
  - Agreed FG definitions are heuristic and non-additive; reframed them as reasoning anchors rather than absolute determinants.
  - Clarified treatment of non-FG edits (heteroatoms retained, alkanes aggregated, ambiguous cases filtered).
  - Explained current system distinguishes FG edits but not purely positional isomers without FG changes.
- Effect: Reviewer was satisfied with clarifications, confirmed positive assessment, and maintained accept.

Ethics reviewer
- Concern: Licensing acknowledgment for MoleculeNet-derived data.
- Author response: Confirmed MIT license attribution would be included in the release.
- Effect: No ethical blockers remained.